# One-Step Distributional Reinforcement Learning

**Mastane Achab**                                                                *mastane.achab@tii.ae*
**Reda Alami, Yasser Abdelaziz Dahou Djilali, Kirill Fedyanin**
*Technology Innovation Institute, 9639 Masdar City, Abu Dhabi, United Arab Emirates*

**Eric Moulines**
*Ecole polytechnique, Palaiseau, France*

*Reviewed on OpenReview:* *https://openreview.net/forum?id=ZPMf53vE1L*

## Abstract

Reinforcement learning (RL) allows an agent interacting sequentially with an environment to maximize its long-term expected return. In the distributional RL (DistrRL) paradigm, the agent goes beyond the limit of the expected value, to capture the underlying probability distribution of the return *across all time steps*. The set of DistrRL algorithms has led to improved empirical performance. Nevertheless, the theory of DistrRL is still not fully understood, especially in the control case. In this paper, we present the simpler *one-step distributional reinforcement learning* (OS-DistrRL) framework encompassing only the randomness induced by the *one-step dynamics* of the environment. Contrary to DistrRL, we show that our approach comes with a unified theory for both policy evaluation and control. Indeed, we propose two OS-DistrRL algorithms for which we provide an almost sure convergence analysis. The proposed approach compares favorably with categorical DistrRL on various environments.

## 1 Introduction

In reinforcement learning (RL), a decision-maker or *agent* sequentially interacts with an unknown and uncertain environment in order to optimize some performance criterion (Sutton & Barto, 2018). At each time step, the agent observes the current state of the environment, then takes an action that influences both its immediate reward and the next state. In other words, RL refers to learning through trial-and-error a strategy (or *policy*) mapping states to actions that maximizes the *long-term cumulative reward* (or *return*): this is the so-called control task. More accurately, this return is a random variable – due to the random transitions across states – and classical RL only focuses on its expected value. On the other hand, the policy evaluation task aims at assessing the quality of any given policy (not necessarily optimal as in control) by computing its expected return in each initial state, also called *value function*. For both evaluation and control, when the *model* (i.e. reward function and transition probabilities between states) is known, these value functions can be seen as fixed points of some operators and computed by dynamic programming (DP) under the Markov decision process (MDP) formalism (Puterman, 2014). Nevertheless, the model in RL is typically unknown and the agent can only approximate the DP approach based on empirical trajectories. The TD(0) algorithm for policy evaluation and Q-learning for control, respectively introduced in (Sutton, 1988) and (Watkins & Dayan, 1992), are flagship examples of the RL paradigm. Formal convergence guarantees were provided for both of these methods, see (Dayan, 1992; Tsitsiklis, 1994; Jaakkola et al., 1993). In many RL applications, the number of states is very large and thus prevents the use of the aforementioned tabular RL algorithms. In such a situation, one should rather use function approximation to approximate the value functions, as achieved by the DQN algorithm (Mnih et al., 2013; 2015) combining ideas from Q-learning and deep learning. More recently, the distributional reinforcement learning (DistrRL) framework was proposed by Bellemare et al. (2017); see also (Morimura et al., 2010a;b). In DistrRL, the agent is optimized to model the whole probability distribution of the return, not just its expectation. In this new paradigm, several distributional procedures where proposed as extensions of the classic (non-distributional)

RL methods, leading to improved empirical performance. In most cases, a DistrRL algorithm is composed of two main ingredients: i) a parametric family of distributions serving as proxies for the distributional returns, and ii) a metric measuring approximation errors between original distributions and parametric proxies.

**Related work.** We recall a few existing DistrRL approaches, all of which considering mixtures of Dirac measures as their parametric family of proxy distributions. Indeed, the Categorical DistrRL (CDRL) parametrization used in the C51 algorithm proposed in (Bellemare et al., 2017) was shown in (Rowland et al., 2018) to correspond to orthogonal projections derived from the Cramér distance[1]. The CDRL approach learns the categorical probabilities $p_1(x, a), \ldots, p_K(x, a)$ of a distribution $\sum_{k=1}^{K} p_k(x, a)\delta_{z_k}$ with fixed predefined support values $z_1, \ldots, z_K$. On the other hand, the quantile regression approach was proposed in (Dabney et al., 2018), where the support $\{Q_1(x, a), \ldots, Q_N(x, a)\}$ of the proxy distribution $\frac{1}{N}\sum_{i=1}^{N}\delta_{Q_i(x,a)}$ is learned for fixed uniform probabilities $\frac{1}{N}$ over the $N$ atoms. In (Dabney et al., 2018), these atoms result from Wasserstein-1 projections, and correspond to quantiles of the unprojected distribution. Achab & Neu (2021) and Achab et al. (2022) later investigated the Wasserstein-2 setup leading to the study of conditional value-at-risk measures. For policy evaluation, the convergence analysis of CDRL and of the quantile approach were respectively derived in (Rowland et al., 2018) and (Rowland et al., 2023). Nevertheless, the theory of DistrRL is more challenging in the control case because the corresponding operator is not a contraction[2] and does not necessarily admits a fixed point. For that reason, Rowland et al. (2018) proved the convergence of CDRL for control only under the restrictive assumption that the optimal policy is unique. Hence, it seems natural to ask the following question: "Is there another formulation of DistrRL in which both the evaluation and the control tasks lead to contractive operators?". The answer provided by this paper is "Yes, via a one-step approach!". Contrary to classic DistrRL dealing with the randomness across all time steps, the one-step variant (originally proposed by Achab (2020)) only cares about the one-step dynamics of the environment.

**Contributions.** Our main contributions are as follows:

- We introduce a one-step variant of the DistrRL framework: we call that approach *one-step distributional reinforcement learning* (OS-DistrRL).

- We show that our new method solves the well-known instability issue of DistrRL in the control case.

- We provide a unified almost-sure convergence analysis for both evaluation and control.

- We experimentally show the competitive performance of our new (deep learning-enhanced) algorithms in various environments.

The paper is organized as follows. In Section 2, we recall a few standard RL tools and notations as well as their DistrRL generalization. Then, our one-step approach is defined in Section 3. Section 4 introduces new OS-DistrRL algorithms along with theoretical convergence guarantees. Finally, numerical experiments are provided for illustration purpose in Section 5. The main proofs are deferred to the Supplementary Material.

**Notations.** The indicator function of any event $E$ is denoted by $\mathbb{I}\{E\}$. We let $\mathcal{P}_b(\mathbb{R})$ be the set of probability measures on $\mathbb{R}$ having bounded support, and $\mathcal{P}(\mathcal{E})$ the set of probability mass functions on any finite set $\mathcal{E}$, whose cardinality is denoted by $|\mathcal{E}|$. The support of any discrete distribution $q \in \mathcal{P}(\mathcal{E})$ is $\text{support}(q) = \{y \in \mathcal{E} : q(y) > 0\}$; the supremum norm of any function $h : \mathcal{E} \to \mathbb{R}$ is $\|h\|_\infty = \max_{y \in \mathcal{E}} |h(y)|$. The cumulative distribution function (CDF) of a probability measure $\nu \in \mathcal{P}_b(\mathbb{R})$ is the mapping $F(z) = \mathbb{P}_{Z \sim \nu}(Z \leq z)$ ($\forall z \in \mathbb{R}$), and we denote its generalized inverse distribution function (a.k.a. quantile function) by $F^{-1} : \tau \in (0, 1] \mapsto \inf\{z \in \mathbb{R}, F(z) \geq \tau\}$. Given $\nu_1, \nu_2 \in \mathcal{P}_b(\mathbb{R})$ with respective CDFs $F_1, F_2$, we denote $\nu_2 \leq \nu_1$ and say that $\nu_1$ *stochastically dominates* $\nu_2$ if $F_1(z) \leq F_2(z)$ for all $z \in \mathbb{R}$. For any probability

---

[1]The Cramér distance $\ell_2$ between two probability distributions $\nu_1, \nu_2$ with CDFs $F_1, F_2$ is equal to $\ell_2(\nu_1, \nu_2) = \sqrt{\int_{\mathbb{R}}(F_1(x) - F_2(x))^2 dx}$ .

[2]A function mapping a metric space to itself is called a $\gamma$-contraction if it is Lipschitz continuous with Lipschitz constant $\gamma < 1$.

measure $\nu \in \mathcal{P}_b(\mathbb{R})$ and measurable function $f \colon \mathbb{R} \to \mathbb{R}$, the *pushforward measure* $f_\# \nu$ is defined for any Borel set $A \subseteq \mathbb{R}$ by $f_\# \nu(A) = \nu(f^{-1}(A)) = \nu(\{z \in \mathbb{R} \colon f(z) \in A\})$. In this article, we only need the affine case $f_{r_0, \gamma}(z) = r_0 + \gamma z$ (with $r_0 \in \mathbb{R}, \gamma \in [0, 1)$) for which $f_{r_0, \gamma \#} \nu \in \mathcal{P}_b(\mathbb{R})$ and $f_{r_0, \gamma \#} \nu(A) = \nu(\{\frac{z - r_0}{\gamma} \colon z \in A\})$ if $\gamma \neq 0$, or $f_{r_0, \gamma \#} \nu = \delta_{r_0}$ is the Dirac measure at $r_0$ if $\gamma = 0$.

## 2 Background on distributional reinforcement learning

In this section, we recall some standard notations, tools and algorithms used in RL and DistrRL.

### 2.1 Markov decision process

Throughout the paper, we consider a Markov decision process (MDP) characterized by the tuple $(\mathcal{X}, \mathcal{A}, P, r, \gamma)$ with finite state space $\mathcal{X}$, finite action space $\mathcal{A}$, transition kernel $P \colon \mathcal{X} \times \mathcal{A} \to \mathcal{P}(\mathcal{X})$, reward function $r \colon \mathcal{X} \times \mathcal{A} \times \mathcal{X} \to \mathbb{R}$, and discount factor $0 \leq \gamma < 1$. If the agent takes some action $a \in \mathcal{A}$ while the environment is in state $x \in \mathcal{X}$, then the next state $X_1$ is sampled from the distribution $P(\cdot|x, a)$ and the immediate reward is equal to $r(x, a, X_1)$. In the discounted MDP setting, the agent seeks a policy $\pi \colon \mathcal{X} \to \mathcal{P}(\mathcal{A})$ maximizing its expected long-term return for each pair $(x, a)$ of initial state and action:

$$Q^\pi(x, a) = \mathbb{E}\left[\sum_{t=0}^{\infty} \gamma^t r(X_t, A_t, X_{t+1}) \,\Big|\, X_0 = x, A_0 = a\right],$$

where $X_{t+1} \sim P(\cdot|X_t, A_t)$ and $A_{t+1} \sim \pi(\cdot|X_{t+1})$. $Q^\pi(x, a)$ and $V^\pi(x) = \sum_{a \in \mathcal{A}} \pi(a|x) Q^\pi(x, a)$ are respectively called the state-action value function and the value function of the policy $\pi$. Further, each of these functions can be seen as the unique fixed point of a so-called Bellman operator (Bellman, 1966), that we denote by $T^\pi$ for Q-functions. For any $Q \colon \mathcal{X} \times \mathcal{A} \to \mathbb{R}$, the image of $Q$ by $T^\pi$ is another Q-function given by:

$$(T^\pi Q)(x, a) = \sum_{x'} P(x'|x, a)\big[r(x, a, x') + \gamma \sum_{a'} \pi(a'|x') Q(x', a')\big].$$

This Bellman operator $T^\pi$ has several nice properties: in particular, it is a $\gamma$-contraction in $\|\cdot\|_\infty$ and thus admits a unique fixed point (by Banach's fixed point theorem), namely $Q^\pi = T^\pi Q^\pi$. It is also well-known from (Bellman, 1966) that there always exists at least one policy $\pi^*$ that is optimal uniformly for all initial conditions $(x, a)$:

$$Q^*(x, a) := Q^{\pi^*}(x, a) = \sup_\pi Q^\pi(x, a) \quad \text{and} \quad V^*(x) := \max_a Q^*(x, a) = V^{\pi^*}(x) = \sup_\pi V^\pi(x).$$

Similarly, this optimal Q-function $Q^*$ is the unique fixed point of some operator $T$ called the Bellman optimality operator and defined by:

$$(TQ)(x, a) = \sum_{x'} P(x'|x, a)\big[r(x, a, x') + \gamma \max_{a'} Q(x', a')\big],$$

which is also a $\gamma$-contraction in $\|\cdot\|_\infty$. Noteworthy, knowing $Q^*$ is sufficient to behave optimally: indeed, a policy $\pi^*$ is optimal if and only if in every state $x$, $\text{support}\big(\pi^*(\cdot|x)\big) \subseteq \arg\max_a Q^*(x, a)$.

### 2.2 The distributional Bellman operator

In distributional RL, we replace scalar-valued functions $Q$ by functions $\mu$ taking values that are entire probability distributions: $\mu^{(x,a)} \in \mathcal{P}_b(\mathbb{R})$ for each pair $(x, a)$. In other words, $\mu$ is a collection of distributions indexed by states and actions. We recall below the definition of the distributional Bellman operator, which generalizes the Bellman operator to distributions.

**Definition 2.1** (DISTRIBUTIONAL BELLMAN OPERATOR, BELLEMARE ET AL. (2017)). Let $\pi$ be a policy. The distributional Bellman operator $\mathcal{T}^\pi \colon \mathcal{P}_b(\mathbb{R})^{\mathcal{X} \times \mathcal{A}} \to \mathcal{P}_b(\mathbb{R})^{\mathcal{X} \times \mathcal{A}}$ is defined for any distribution function $\mu = (\mu^{(x,a)})_{x,a}$ by

$$(\mathcal{T}^\pi \mu)^{(x,a)} = \sum_{x',a'} P(x'|x, a)\pi(a'|x') f_{r(x,a,x'),\gamma \#} \mu^{(x',a')}.$$

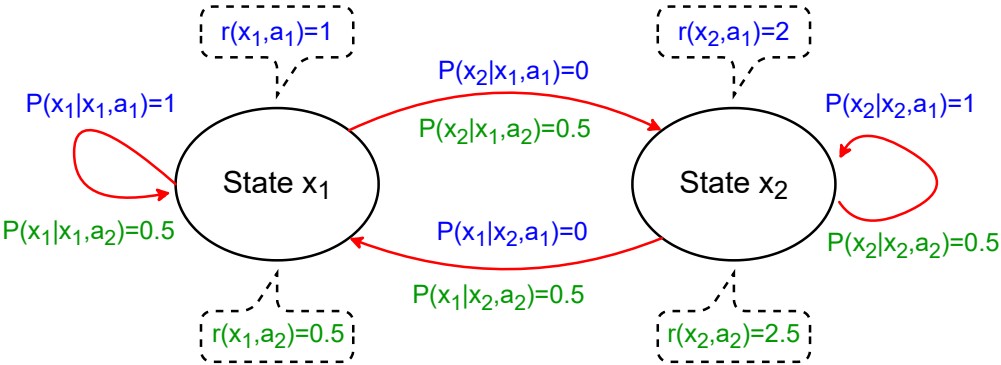

Figure 1: Markov decision process with two states $\mathcal{X} = \{x_1, x_2\}$, two actions $\mathcal{A} = \{a_1, a_2\}$ and reward function independent of the next state: $r(x, a, \cdot) \equiv r(x, a)$.

We know from Bellemare et al. (2017) that $\mathcal{T}^\pi$ is a $\gamma$-contraction in the maximal $p$-Wasserstein metric[3]

$$\overline{W}_p(\mu_1, \mu_2) = \max_{(x,a) \in \mathcal{X} \times \mathcal{A}} W_p(\mu_1^{(x,a)}, \mu_2^{(x,a)})$$

at any order $p \in [1, +\infty]$. Consequently, $\mathcal{T}^\pi$ has a unique fixed point $\mu_\pi = (\mu_\pi^{(x,a)})_{x,a}$ equal to the collection of the probability distributions of the returns:

$$\mu_\pi^{(x,a)} = \text{Distr}\left(\sum_{t=0}^{\infty} \gamma^t r(X_t, A_t, X_{t+1}) \,\middle|\, X_0 = x, A_0 = a\right). \tag{1}$$

Applying $\mathcal{T}^\pi$ may be prohibitive in terms of space complexity: if $\mu$ is discrete, then $\mathcal{T}^\pi \mu$ is still discrete but with up to $|\mathcal{X}| \cdot |\mathcal{A}|$ times more atoms, as shown in the following example.

**Example 2.2.** Let $\mu = (\mu^{(x,a)})_{x,a}$ be the collection of the atomic distributions

$$\mu^{(x,a)} = \sum_{k=1}^{K} p_k(x, a) \delta_{Q_k(x,a)},$$

where $K \geq 1$, $p_k(x, a) \geq 0$ and $p_1(x, a) + \cdots + p_K(x, a) = 1$. Then, $\mathcal{T}^\pi \mu$ is also a collection of atomic distributions with (at most) $|\mathcal{X}| \cdot |\mathcal{A}|$ times more atoms:

$$(\mathcal{T}^\pi \mu)^{(x,a)} = \sum_{x',a'} P(x'|x,a)\pi(a'|x') \sum_{k=1}^{K} p_k(x', a')\delta_{r(x,a,x')+\gamma Q_k(x',a')},$$

where we stress that the entire expression $[r(x, a, x') + \gamma Q_k(x', a')]$ is the argument of the Dirac delta function.

**Projected operators.** Motivated by this space complexity issue, projected DistrRL operators were proposed to ensure a predefined and fixed space complexity budget. A projected DistrRL operator is the composition of a DistrRL operator with a projection over some parametric family of distributions. The Cramér distance projection $\Pi_{\mathcal{C}}$ has been considered in (Bellemare et al., 2017; Rowland et al., 2018; Bellemare et al., 2019): we recall its definition below.

**Definition 2.3** (CRAMÉR PROJECTION). Let $K \geq 2$ and $z_1 < \cdots < z_K$ be real numbers defining the support of the categorical distributions. The Cramér projection is then defined by: for all $z' \in \mathbb{R}$,

$$\Pi_{\mathcal{C}}(\delta_{z'}) = \begin{cases} \delta_{z_1} & \text{if } z' \leq z_1 \\ \frac{z_{j+1}-z'}{z_{j+1}-z_j}\delta_{z_j} + \frac{z'-z_j}{z_{j+1}-z_j}\delta_{z_{j+1}} & \text{if } z_j < z' \leq z_{j+1} \\ \delta_{z_K} & \text{if } z' > z_K \end{cases}$$

---

[3]For $p \geq 1$, we recall that the $p$-Wasserstein distance between two probability distributions $\nu_1, \nu_2$ on $\mathbb{R}$ with CDFs $F_1, F_2$ is defined as $W_p(\nu_1, \nu_2) = \left(\int_{\tau=0}^{1} \left|F_1^{-1}(\tau) - F_2^{-1}(\tau)\right|^p d\tau\right)^{\frac{1}{p}}$. If $p = \infty$, $W_\infty(\nu_1, \nu_2) = \sup_{\tau \in (0,1)} |F_1^{-1}(\tau) - F_2^{-1}(\tau)|$.

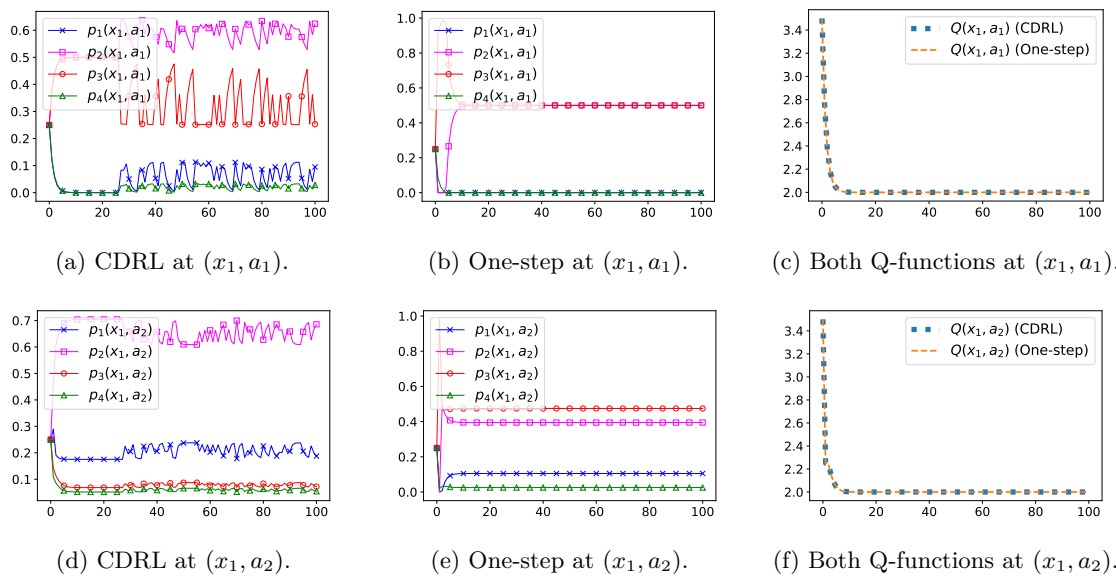

(a) CDRL at $(x_1, a_1)$.  (b) One-step at $(x_1, a_1)$.  (c) Both Q-functions at $(x_1, a_1)$.

(d) CDRL at $(x_1, a_2)$.  (e) One-step at $(x_1, a_2)$.  (f) Both Q-functions at $(x_1, a_2)$.

Figure 2: Instability of categorical distributional control compared to the convergence of our one-step approach. The two leftmost plots are the iterations of the Cramér-projected "full" optimality operator $\Pi_{\mathcal{C}}\mathcal{T}$, while the center plots are obtained with our one-step operator $\Pi_{\mathcal{C}}\mathcal{T}$. The rightmost plots correspond to the Q-functions $Q(x, a) = \sum_{k=1}^{4} p_k(x, a) z_k$ .

and extended linearly to mixtures of Dirac measures.

We will also apply $\Pi_{\mathcal{C}}$ entrywise to collections of distributions: if $\mu = (\mu^{(x,a)})_{x,a}$, then $\Pi_{\mathcal{C}}\mu = (\Pi_{\mathcal{C}}\mu^{(x,a)})_{x,a}$. The Cramér projection satisfies an important *mean-preserving property*: for any discrete distribution $q$ with support included in the interval $[z_1, z_K]$, then:

$$\mathbb{E}_{Z \sim \Pi_{\mathcal{C}}(q)}[Z] = \mathbb{E}_{Z \sim q}[Z] . \tag{2}$$

In this line of work, the proxy distributions $\sum_{k=1}^{K} p_k(x, a) \delta_{z_k}$ are parametrized by the categorical probabilities $p_1(x, a), \ldots, p_K(x, a)$. In (Dabney et al., 2018), the Wasserstein-1 projection $\Pi_{W_1}$ over atomic distributions $\frac{1}{N} \sum_{i=1}^{N} \delta_{Q_i(x,a)}$ is used. For this specific projection, the atoms that best approximate some CDF $F_{x,a}$ are obtained through quantile regression with $Q_i(x, a) = F_{x,a}^{-1}(\frac{2i-1}{2N})$.

## 2.3 Instability of distributional control

Bellemare et al. (2017) have also proposed DistrRL optimality operators $\mathcal{T}$, defined such that $\mathcal{T}\mu = \mathcal{T}^{\pi_G}\mu$ where $\pi_G$ is a greedy policy with respect to the expectation of $\mu$. The authors showed in their Propositions 1-2 that, unfortunately, $\mathcal{T}$ is not a contraction and does not necessarily admits a fixed point. For that reason, the convergence analysis of the control case in DistRL is more challenging than for the evaluation task. Rowland et al. (2018) and Bellemare et al. (2023) (see chapter 7.4, in particular Theorem 7.9) circumvent this issue by reducing the control task to evaluation under the assumption that the optimal policy is unique. We start our investigation by verifying that this uniqueness hypothesis is critical to ensure convergence in DistRL control. For that purpose, we choose a toy example – fully described in Section 5 – with (infinitely) many optimal policies and we run a CDRL procedure over it. As expected, we observe in Figure 2 the unstable behavior of the DistRL paradigm for the control task. Indeed, as explained in chapter 7.5 in Bellemare et al. (2023), multiple optimal policies can produce different distributions that are then mixed by DistRL control algorithms which might never converge. In the next section, we introduce our new "one-step" distributional approach converging even in this situation since all optimal policies will be shown to share the same fixed point distribution.

## 3 One-step distributional operators

This section introduces the building blocks of our new framework. The main goal here is to show that our one-step approach is theoretically sound, though being less ambitious than "full" DistrRL. The main benefit of this simplification is that it will allow us to derive in the next section a convergence result holding for both evaluation and control. In particular in the control case, we will *not* need any additional assumption such as the uniqueness of the optimal policy as in CDRL (Rowland et al., 2018).

### 3.1 Formal definitions

Let us now define the one-step DistrRL operators. Intuitively, they are similar to the "full" DistrRL operator $\mathcal{T}^\pi$ except that they average the randomness after the first random transition and are thus oblivious to the randomness induced by the remaining time steps.

**Definition 3.1** (ONE-STEP DISTRIBUTIONAL BELLMAN OPERATOR). Let $\pi$ be a policy. The one-step distributional Bellman operator $\mathcal{T}^\pi \colon \mathcal{P}_b(\mathbb{R})^{\mathcal{X} \times \mathcal{A}} \to \mathcal{P}_b(\mathbb{R})^{\mathcal{X} \times \mathcal{A}}$ is defined for any distribution function $\mu$ by

$$(\mathcal{T}^\pi \mu)^{(x,a)} = \sum_{x'} P(x'|x,a) \delta_{r(x,a,x') + \gamma \sum_{a'} \pi(a'|x') \mathbb{E}_{Z \sim \mu^{(x',a')}}[Z]} ,$$

where we stress that the argument of the Dirac delta function is: $r(x,a,x') + \gamma \sum_{a'} \pi(a'|x') \mathbb{E}_{Z \sim \mu^{(x',a')}}[Z]$.

**Definition 3.2** (ONE-STEP DISTRIBUTIONAL BELLMAN OPTIMALITY OPERATOR). The one-step distributional Bellman optimality operator $\mathcal{T} \colon \mathcal{P}_b(\mathbb{R})^{\mathcal{X} \times \mathcal{A}} \to \mathcal{P}_b(\mathbb{R})^{\mathcal{X} \times \mathcal{A}}$ is defined for any $\mu$ by

$$(\mathcal{T} \mu)^{(x,a)} = \sum_{x'} P(x'|x,a) \delta_{r(x,a,x') + \gamma \max_{a'} \mathbb{E}_{Z \sim \mu^{(x',a')}}[Z]} .$$

**Example 3.3.** Let $\mu$ be the same discrete distribution function as in Example 2.2. Then,

$$(\mathcal{T}^\pi \mu)^{(x,a)} = \sum_{x'} P(x'|x,a) \delta_{r(x,a,x') + \gamma \sum_{a'} \pi(a'|x') Q(x',a')}$$

$$\text{and} \qquad (\mathcal{T} \mu)^{(x,a)} = \sum_{x'} P(x'|x,a) \delta_{r(x,a,x') + \gamma \max_{a'} Q(x',a')} ,$$

where $Q(x',a') = \sum_{k=1}^K p_k(x',a') Q_k(x',a')$ .

Similarly to $\mathcal{T}$, the one-step operators $\mathcal{T}$ and $\mathcal{T}^\pi$ lead to a space complexity issue by producing distributions with a number $|\mathcal{X}|$ of atoms, which can be too demanding for a large state space.

### 3.2 Main properties

Let us now discuss some key properties satisfied by our new one-step distributional operators. We first show that, similarly to the non-distributional setting, our approach comes with contractive operators in both evaluation and control.

**Proposition 3.1** (CONTRACTIVITY). *Let $\pi$ be a policy.*

*(i) For any $1 \leq p \leq \infty$, the one-step operators $\mathcal{T}^\pi$ and $\mathcal{T}$ are $\gamma$-contractions in $\overline{W}_p$.*

*(ii) The Cramér-projected one-step operators $\Pi_{\mathcal{C}} \circ \mathcal{T}^\pi$ and $\Pi_{\mathcal{C}} \circ \mathcal{T}$ are $\gamma$-contractions in $\overline{W}_1$.*

We stress that Proposition 3.1 highly contrasts with classic DistrRL where the control operator $\mathcal{T}$ is not a contraction in any metric. A major consequence of contractivity is the existence and uniqueness of fixed points, whose explicit formulas are gathered in the next proposition and in Table 1.

**Proposition 3.2** (FIXED POINTS). *Let $\pi$ be a policy and consider $\Pi_{\mathcal{C}}$ from Definition 2.3.*

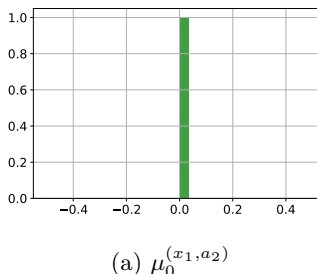

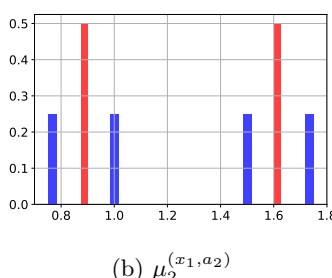

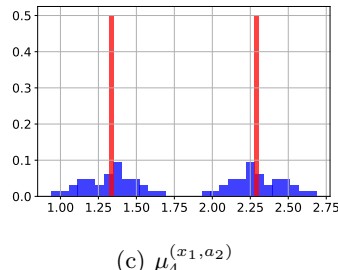

(a) $\mu_0^{(x_1,a_2)}$

(b) $\mu_2^{(x_1,a_2)}$

(c) $\mu_4^{(x_1,a_2)}$

Figure 3: Illustration of Examples 2.2-3.3 in the two-states two-actions MDP depicted in Figure 1, with discount factor $\gamma = \frac{1}{2}$, for the stochastic policy $\pi(a|x) = \frac{1}{2}$ for all $x, a$. The probability distribution $\mu_j^{(x_1,a_2)}$ with histogram represented in red (resp. in blue) is obtained by applying $j$ successive times the one-step distributional operator $\mathfrak{T}^\pi$ (resp. the "full" distributional operator $\mathcal{T}^\pi$) to the initial distributions $\mu_0^{(x,a)} = \delta_0$. The red "one-step" histogram has a constant number of 2 atoms, corresponding to the number of states. Meanwhile, the blue "full" histogram approximates a continuous probability density with an increasing number of atoms along iterations.

(i) *The unique fixed point of $\mathfrak{T}^\pi$ is $\nu_\pi$ given by*

$$\nu_\pi^{(x,a)} = \sum_{x'} P(x'|x,a) \delta_{r(x,a,x') + \gamma V^\pi(x')} \,.$$

(ii) *The unique fixed point of $\mathfrak{T}$ is $\nu_*$ given by*

$$\nu_*^{(x,a)} = \sum_{x'} P(x'|x,a) \delta_{r(x,a,x') + \gamma V^*(x')} \,.$$

(iii) *If $z_1 \le r(x,a,x') + \gamma V^\pi(x') \le z_K$ for all triplets $(x,a,x')$, then the unique fixed point of $\Pi_\mathcal{C} \circ \mathfrak{T}^\pi$ is $\eta_\pi = \Pi_\mathcal{C}(\nu_\pi)$.*

(iv) *If $z_1 \le r(x,a,x') + \gamma V^*(x') \le z_K$ for all triplets $(x,a,x')$, then the unique fixed point of $\Pi_\mathcal{C} \circ \mathfrak{T}$ is $\eta_* = \Pi_\mathcal{C}(\nu_*)$.*

The proof of Proposition 3.2 (point i, for instance) involves replacing, inside each Dirac, the quantity $\sum_{x''} P(x''|x',a')(r(x',a',x'') + \gamma V^\pi(x''))$ with the simpler Q-value $Q^\pi(x',a')$. This substitution is permissible due to the unique fixed-point characterization of $V^\pi$ and $Q^\pi$. Interestingly, Proposition 3.2-(iii)-(iv) shows that, in our one-step framework, the fixed point of a projected operator is simply the projection of the fixed point of the unprojected operator. Although this fact seems natural, it is not necessarily true in DistrRL. Indeed, the proof of Proposition 3 in (Rowland et al., 2018) suggests that the fixed point of $\Pi_\mathcal{C} \circ \mathcal{T}^\pi$ is a worse approximation of $\mu_\pi$ than is $\Pi_\mathcal{C}\mu_\pi$, by a multiplicative factor $\sqrt{1/(1-\gamma)}$ in terms of Cramér distance. Furthermore, we stress that two policies sharing the same value function but with potentially distinct distributions and risk-levels (e.g. different variances) cannot be distinguished via the one-step approach as, from Proposition 3.2-(i), they necessarily share the same fixed point. This observation highlights a notable limitation of one-step DistrRL: it is unable to distinguish between two policies with identical value functions that would, however, yield distinct fixed points in the context of full DistrRL. Equipped with our projected one-step operators, we derive in the next section variants of the CDRL algorithms.

## 4 One-step DistrRL algorithms

This section introduces new categorical algorithms based on the Cramér projection together with formal convergence guarantees.

|  | DistrRL | One-step DistrRL |
|---|---|---|
| Evaluation | Distr $\left(\sum_{t=0}^{\infty} \gamma^t r(X_t, A_t, X_{t+1})\right)$ | $\sum_{x'} P(x'\|x,a)\delta_{r(x,a,x')+\gamma V^\pi(x')}$ |
| Control | does not necessarily exist | $\sum_{x'} P(x'\|x,a)\delta_{r(x,a,x')+\gamma V^*(x')}$ |

Table 1: Comparison of the fixed points of the distributional Bellman operators in DistrRL versus one-step DistrRL.

|  | CDRL | One-step CDRL |
|---|---|---|
| Categorical target | $\Pi_{\mathcal{C}}\left(\sum_{k=1}^{K} p_{t,k}(x_{t+1},a^*)\delta_{r_t+\gamma z_k}\right)$ | $\Pi_{\mathcal{C}}\left(\delta_{r_t+\gamma Q_t(x_{t+1},a^*)}\right)$ |
| Time complexity | $\mathcal{O}(K\log K)$ | $\mathcal{O}(\log K)$ |

Table 2: Comparison of the categorical targets in CDRL versus one-step CDRL. The notation $a^*$ refers to a greedy action, namely $a^* \in \arg\max_{a'} Q_t(x_{t+1}, a')$.

## 4.1 One-Step CDRL

We propose two algorithms, for policy evaluation and control respectively, that are described in Algorithm 1. These new categorical methods are derived from the stochastic approximation of the projected operators $\Pi_{\mathcal{C}} \circ \mathcal{T}^\pi$ and $\Pi_{\mathcal{C}} \circ \mathcal{T}$. For each state-action pair $(x, a)$, both methods learn a discrete probability distribution $p_1(x, a), \ldots, p_K(x, a)$ over some fixed support $z_1 < \cdots < z_K$. As in classic RL algorithms, we consider a sequence of stepsizes $\alpha_t(x, a) \geq 0$ indexed by states, actions and time steps $t \geq 0$. At each time $t$, we perform a mixture update between the current distribution and a distributional target which is the Cramér projection of a single Dirac mass located at the target of TD(0) or Q-learning, computed from a single transition $(x_t, a_t, r_t, x_{t+1})$. The only difference with tabular CDRL lies in the target as shown in Table 2. Contrary to the original CDRL target, ours remains the same whatever the greedy action $a^*$ we choose inside the set $\arg\max_{a'} Q_t(x_{t+1}, a')$: this explains why our approach is stable even when the optimal policy is not unique as illustrated in Figure 2. Moreover, our categorical target is faster to compute than in CDRL, where the time complexity $\mathcal{O}(K\log(K))$ pays the price for inserting every atom $r_t + \gamma z_k$ into the sorted array $(z_1, \ldots, z_K)$. Before moving forward to the convergence analysis of Algorithm 1, we propose its deep

---

**Algorithm 1** Tabular one-step categorical DistrRL

---

**Input:** $\eta_t^{(x,a)} = \sum_{k=1}^{K} p_{t,k}(x,a)\delta_{z_k}$ for all $(x,a)$
  Sample transition: $(x_t, a_t, r_t, x_{t+1})$
  Estimate Q-values: $Q_t(x_{t+1}, a) \leftarrow \sum_{k=1}^{K} p_{t,k}(x_{t+1}, a) \cdot z_k$
  **if** policy evaluation **then**
    $\widehat{\eta}_t^{(x_t,a_t)} \leftarrow \Pi_{\mathcal{C}}(\delta_{r_t+\gamma \sum_{a'} \pi(a'\|x_{t+1})Q_t(x_{t+1},a')}) = \sum_{k=1}^{K} \widehat{p}_{t,k}\delta_{z_k}$
  **else if** control **then**
    $\widehat{\eta}_t^{(x_t,a_t)} \leftarrow \Pi_{\mathcal{C}}(\delta_{r_t+\gamma \max_{a'} Q_t(x_{t+1},a')}) = \sum_{k=1}^{K} \widehat{p}_{t,k}\delta_{z_k}$
  **end if**
  Mixture update: $\eta_{t+1}^{(x_t,a_t)} \leftarrow (1 - \alpha_t(x_t, a_t))\eta_t^{(x_t,a_t)} + \alpha_t(x_t, a_t)\widehat{\eta}_t^{(x_t,a_t)}$
  $\eta_{t+1}^{(x,a)} \leftarrow \eta_t^{(x,a)}$ , $\forall(x,a) \neq (x_t, a_t)$
**Output:** $\eta_{t+1}$

---

RL counterpart in Algorithm 2 based on the minimization of a Kullback-Leibler (KL) loss.

**Deep one-step CDRL.** The main challenge in a non-tabular context is to learn the distributions in a compact and efficient way. For that purpose, we use the same deep categorical approach as in C51 (Bellemare et al., 2017). As shown in Table 1, the one-step method aims at learning much simpler distributions (necessarily atomic) than full DistrRL (typically, continuous distributions carrying more information). This suggests choosing a smaller number of categories, e.g. $K = 4$ used in Section 5, compared to what is commonly used in CDRL. Due to its similarity with C51, we call this new algorithm "OS-C51".

---

**Algorithm 2** OS-C51 (single update)

---

**Input:**   categorical distributions $\eta_\theta^{(x,a)} = \sum_{k=1}^K p_{\theta,k}(x,a)\delta_{z_k}$ and a transition $(x_t, a_t, r_t, x_{t+1})$

Compute Q-function in next state: $Q(x_{t+1}, a') \leftarrow \sum_{k=1}^K p_{\theta,k}(x_{t+1}, a')z_k$

Compute categorical target: $\widehat{\eta}^{(x_t,a_t)} \leftarrow \Pi_{\mathcal{C}}(\delta_{r_t + \gamma \max_{a'} Q(x_{t+1}, a')})$

**Output:**   $\mathrm{KL}(\widehat{\eta}^{(x_t,a_t)} \| \eta_\theta^{(x_t,a_t)})$

---

### 4.2   Convergence analysis

We now provide convergence guarantees for our tabular one-step DistrRL algorithms. The major difference with the analysis of CDRL (Rowland et al., 2018) is that we do not require the uniqueness of the optimal policy in the case of control. We also rely on the existing analysis of non-distributional RL (Dayan, 1992; Tsitsiklis, 1994; Jaakkola et al., 1993). In particular, we require the following standard assumption.

**Assumption 4.1.** For any pair $(x, a) \in \mathcal{X} \times \mathcal{A}$, the stepsizes $(\alpha_t(x, a))_{t \geq 0}$ satisfy the Robbins-Monro conditions:

$$\begin{cases} \sum_{t=0}^\infty \alpha_t(x,a) = \infty \\ \sum_{t=0}^\infty \alpha_t(x,a)^2 < \infty \end{cases} \quad \text{almost surely.}$$

Equipped with stepsizes $(\alpha_t(x, a))$ satisfying the Robbins-Monro conditions, we are now ready to state our main theoretical contribution: namely, the convergence of Algorithm 1.

**Theorem 4.1.** *Consider Algorithm 1 and let us assume that Assumption 4.1 holds for the stepsizes* $(\alpha_t(x, a))_{t,x,a}$ .

*(i) In the case of evaluation of a policy* $\pi$,

$$\overline{W}_1(\eta_t, \eta_\pi) \xrightarrow{t \to \infty} 0 \quad \text{almost surely},$$

*where* $\eta_\pi$ *is defined in Proposition 3.2-(iii).*

*(ii) In the case of control,*

$$\overline{W}_1(\eta_t, \eta_*) \xrightarrow{t \to \infty} 0 \quad \text{almost surely},$$

*where* $\eta_*$ *is defined in Proposition 3.2-(iv).*

The proof of Theorem 4.1 is deferred to the Supplementary Material: it follows the same steps as the proofs of Theorem 2 in (Tsitsiklis, 1994) and Theorem 1 in (Rowland et al., 2018). Notably, our analysis remains the same for evaluation and control contrary to (Rowland et al., 2018), where the control case requires additional assumptions (namely, uniqueness of the optimal policy and for all $t \geq 0$, for all $1 \leq k \leq K$ such that $p_{t,k}(x_{t+1}, a^*) \neq 0$, $r_t + \gamma z_k \in [z_1, z_K]$ almost surely).

## 5   Numerical experiments

In this section, we present numerical experiments on both tabular and Atari games environments.

### 5.1   Tabular setting

We describe our tabular experiments: first in a dynamic programming context i.e. knowing the transition kernel $P$ and the reward function $r$, then in the "Frozen Lake" environment by only observing empirical transitions.

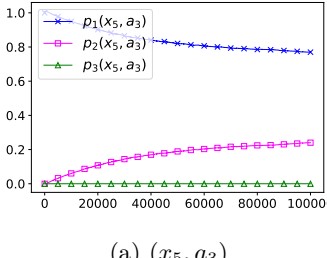 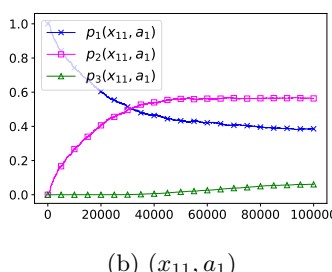 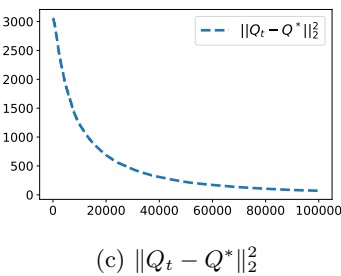

(a) $(x_5, a_3)$       (b) $(x_{11}, a_1)$       (c) $\|Q_t - Q^*\|_2^2$

Figure 4: Learning dynamics of the atomic probability distribution $(p_{t,k}(x,a))_{1 \leq k \leq K=3}$ produced by Algorithm 1 (control) on the Frozen Lake environment across 100000 iterations, with support $(z_1, z_2, z_3) = (0, 10, 20)$, constant stepsize $\alpha = 0.6$ and $\varepsilon$-greedy exploration. The results are averaged over 100 seeds.

**Distributional dynamic programming.** Figures 2-3 are obtained by exact dynamic programming in the MDP in Figure 1 with $\gamma = \frac{1}{2}$: in this specific case, all policies are optimal. As discussed in subsection 2.3, this handcrafted example reveals the instability of classic CDRL in Figure 2 contrary to our one-step approach: for both methods we consider $K = 4$ categories with $(z_1, z_2, z_3, z_4) = (0, 1.9, 2.1, 10)$. We point out that our results are sensitive to this specific choice of support. Indeed, always taking the action $a_1$ in state $x_1$ results in a total discounted distributional return reduced to a Dirac mass at 2. However, taking the action $a_2$ leads to the same expected value $Q^*(x_1, a_2) = 2$ but with non-zero variance. Thus, CDRL targets can occasionally (when taking action $a_2$) fall outside the selected interval $[1.9, 2.1]$, leading to the depicted instability. Figure 3 illustrates the space complexity issue of DistrRL without projection: while the number of atoms is multiplied by at most $|\mathcal{X}||\mathcal{A}|$ after each application of $\mathcal{T}^\pi$, our one-step operators produce distributions with (at most) as many atoms as there are states, i.e. two in this example.

**Frozen Lake.** We also consider the Frozen Lake environment from OpenAI Gym (Brockman et al., 2016) with discount factor $\gamma = 0.95$. It is characterized by 16 states $x_1, \ldots, x_{16}$ and four actions $a_1, \ldots, a_4$. Plus, this is a stochastic environment i.e. the transition probabilities $P(x'|x, a)$ are not all equal to either 0 or 1, which justifies a distributional approach. The FrozenLake environment is a gridworld game (4x4 grid), where an agent moves on slippery ice (stochastic transitions) towards a goal. The agent can take 4 actions (up, down, left, right), aiming to reach the goal state from the starting state, avoiding holes, and only achieving a reward of 1 upon reaching the goal, and 0 otherwise. Due to the slippery attribute of the environment, the transitions are stochastic, meaning the agent may not always end up in the state it intended to move to. For example, if the agent chooses to move right, it may actually move up, right, or down with equal probability $1/3$ due to the slippery ice. In Figure 4, we plot the iterates $p_{t,k}(x, a)$ over 100000 steps generated by our tabular Algorithm 1 with $K = 3$ atoms and $(z_1, z_2, z_3) = (0, 10, 20)$, constant stepsize $\alpha = 0.6$ and $\varepsilon$-greedy exploration with $\varepsilon$ exponentially decaying from 1 to 0.25. We average the results over 100 seeds. Given a pair $(x, a)$, we observe the joint convergence of the three probabilities. As in CDRL, and because of the mean-preserving property (Eq. 2), the average $Q_t(x, a) = \sum_{k=1}^{3} p_{t,k}(x, a) z_k$ coincides with the Q-learning iterates and converges to $Q^*$.

## 5.2 Atari games

For the experiments on Atari games (Bellemare et al., 2013), we implement [4] the OS-C51 agent on top of the popular "CleanRL" codebase (Huang et al., 2022). We compare the OS-C51 algorithm against C51 for two different number of atoms: $K = 4$ or $K = 51$. In each of the two cases, we choose the support to be evenly spread over the interval $[-10, 10]$: $z_1 = -10 < \cdots < z_K = 10$. We use the same architecture as C51: a deep neural network, parameterized by $\theta$, takes an observation as input and outputs a vector of $K$ logits. We run DQN and C51 with the default hyperparameters from CleanRL; for our new OC-C51 method, we take the same hyperparameters as in C51 due to their similarities. All three methods are based on a deep neural network with 3 convolutional layers followed by 2 fully connected layers with ReLU activation functions.

---

[4]here: `https://github.com/mastane/cleanrl`

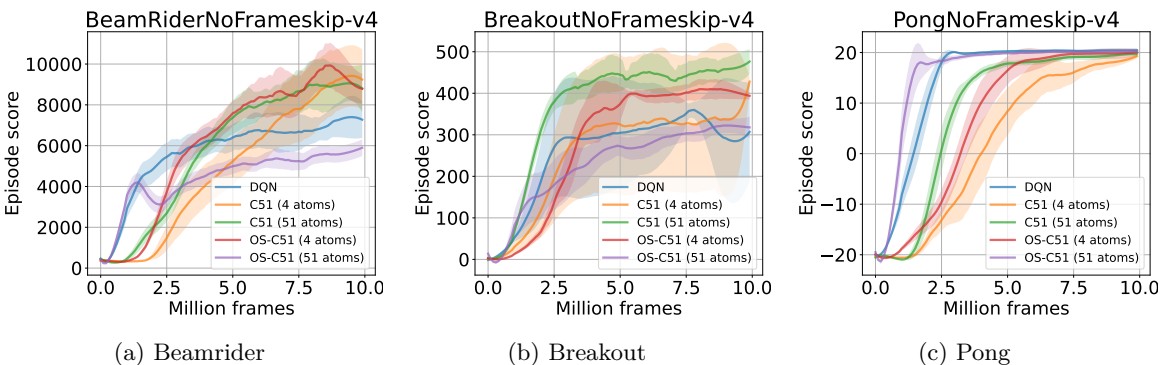

Figure 5: Average evaluation episodic return over 10 million frames for three Atari games.

They all use the same linearly decaying $\varepsilon$-greedy exploration and the transitions are collected in a replay memory buffer of size 1000000. In particular for DQN, we use the Adam optimizer (Kingma & Ba, 2014) with learning rate equal to 0.0001 and batch size of 32. For the optimization of C51 and OS-C51, we use the Adam optimizer with learning rate set to 0.00025 and batch size equal to 32. The results are averaged over 5 seeds. As shown in Figure 5, the performance of OS-C51 (with only four atoms) is comparable with C51 on the Beamrider and Pong games. However, there seems to be a significant advantage for C51 on Breakout: this could be attributed to C51's objective of approximating distributions with more complex and richer information, potentially offering a deeper understanding and better representation of the game dynamics. Finally, we highlight that we did not use "sticky actions", where a chosen action is randomly repeated for several consecutive frames, introducing a form of environmental stochasticity. This could be an interesting source of stochasticity to explore for future experimental investigation, potentially adding complexity and richness to the learning process.

## 6 Conclusion

We proposed new distributional RL algorithms that naturally extend TD(0) and Q-learning in the tabular setting. The main novelty in our approach is the use of new one-step distributional operators circumventing the instability issues of DistrRL control. We provided both theoretical convergence analysis and empirical proof-of-concept. A significant limitation of one-step DistrRL, as underscored in our study, is its inability to distinguish between two policies with identical value functions but differing distributions. This points to a crucial advantage of full DistrRL, which can capture and exploit these subtleties. Future research could investigate the generalization of our method based on the dynamics of several successive steps instead of a single one. Investigating this multi-step extension would likely reveal further important insights into the dynamics of RL processes. Another promising direction for future research lies in exploring the possible connection between full DistrRL and a multi-step approach with infinitely many steps. Such an investigation could offer a deeper understanding of the relationship between these methods and help to clarify the distinct benefits and trade-offs involved.

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

## A  Proof of Proposition 3.1

i). *Contraction in Wasserstein distance.* Let $1 \le p < \infty$. Let $\mu_1, \mu_2 \in \mathcal{P}_b(\mathbb{R})^{\mathcal{X} \times \mathcal{A}}$ be two distribution functions. Denoting $Q_1(x,a)$ and $Q_2(x,a)$ the respective expectations of $\mu_1^{(x,a)}$ and $\mu_2^{(x,a)}$, we have for any pair $(x,a)$:

$$W_p^p((\mathcal{T}\mu_1)^{(x,a)}, (\mathcal{T}\mu_2)^{(x,a)}) =$$
$$W_p^p\Big(\sum_{x'} P(x'|x,a)\delta_{r(x,a,x')+\gamma \max_{a'} Q_1(x',a')}, \sum_{x'} P(x'|x,a)\delta_{r(x,a,x')+\gamma \max_{a'} Q_2(x',a')}\Big)$$
$$\le \sum_{x'} P(x'|x,a) W_p^p\big(\delta_{r(x,a,x')+\gamma \max_{a'} Q_1(x',a')}, \delta_{r(x,a,x')+\gamma \max_{a'} Q_2(x',a')}\big)$$
$$= \gamma^p \sum_{x'} P(x'|x,a) |\max_{a'} Q_1(x',a') - \max_{a'} Q_2(x',a')|^p$$
$$\le \gamma^p \sum_{x'} P(x'|x,a) \max_{a'} |Q_1(x',a') - Q_2(x',a')|^p = \gamma^p \sum_{x'} P(x'|x,a) \max_{a'} \Big| \int_{\tau=0}^{1} (F_{x',a'}^{-1}(\tau) - \tilde{F}_{x',a'}^{-1}(\tau)) d\tau \Big|^p$$
$$\le \gamma^p \sum_{x'} P(x'|x,a) \max_{a'} \underbrace{\int_{\tau=0}^{1} \big| F_{x',a'}^{-1}(\tau) - \tilde{F}_{x',a'}^{-1}(\tau) \big|^p d\tau}_{W_p^p(\mu_1^{(x',a')}, \mu_2^{(x',a')})} \le \gamma^p \overline{W}_p^p(\mu_1, \mu_2),$$

where $F_{x,a}, \tilde{F}_{x,a}$ denote the respective CDFs of $\mu_1^{(x,a)}$ and $\mu_2^{(x,a)}$, and the first inequality follows from the interpretation of the Wasserstein distance as an infimum over couplings. Indeed, we have upper bounded

$$W_p^p((\mathcal{T}\mu_1)^{(x,a)}, (\mathcal{T}\mu_2)^{(x,a)}) = \inf_{\lambda \in \Lambda} \mathbb{E}_{(Z_1,Z_2) \sim \lambda}[|Z_1 - Z_2|^p],$$

where $\Lambda$ denotes the set of all couplings of $(\mathcal{T}\mu_1)^{(x,a)}$ and $(\mathcal{T}\mu_2)^{(x,a)}$, via a specific coupling, $\lambda_0$, such that

$$\lambda_0(\{r(x,a,x') + \gamma \max_{a'} Q_1(x',a')\} \times \{r(x,a,x') + \gamma \max_{a'} Q_2(x',a')\}) = P(x'|x,a), \ \forall x'.$$

Hence, by taking the supremum over all $(x,a)$, we deduce that $\mathcal{T}$ is a $\gamma$-contraction in $\overline{W}_p$:

$$\overline{W}_p(\mathcal{T}\mu_1, \mathcal{T}\mu_2) \le \gamma \overline{W}_p(\mu_1, \mu_2).$$

Similarly for $p = \infty$,

$$W_\infty((\mathfrak{T}\mu_1)^{(x,a)}, (\mathfrak{T}\mu_2)^{(x,a)}) =$$

$$W_\infty(\sum_{x'} P(x'|x,a)\delta_{r(x,a,x')+\gamma\max_{a'}Q_1(x',a')}, \sum_{x'} P(x'|x,a)\delta_{r(x,a,x')+\gamma\max_{a'}Q_2(x',a')})$$

$$\leq \sum_{x'} P(x'|x,a)W_\infty(\delta_{r(x,a,x')+\gamma\max_{a'}Q_1(x',a')}, \delta_{r(x,a,x')+\gamma\max_{a'}Q_2(x',a')})$$

$$= \gamma \sum_{x'} P(x'|x,a)|\max_{a'}Q_1(x',a') - \max_{a'}Q_2(x',a')|$$

$$\leq \gamma \sum_{x'} P(x'|x,a)\max_{a'}|Q_1(x',a') - Q_2(x',a')| = \gamma \sum_{x'} P(x'|x,a)\max_{a'}\left|\int_{\tau=0}^1 (F_{x',a'}^{-1}(\tau) - \tilde{F}_{x',a'}^{-1}(\tau))d\tau\right|$$

$$\leq \gamma \sum_{x'} P(x'|x,a)\max_{a'}\int_{\tau=0}^1 |F_{x',a'}^{-1}(\tau) - \tilde{F}_{x',a'}^{-1}(\tau)|d\tau \leq \gamma\overline{W}_\infty(\mu_1,\mu_2).$$

For policy evaluation, one can follow the same proofs by incorporating the following additional step:

$$\left|\sum_{a'} \pi(a'|x')(Q_1(x',a') - Q_2(x',a'))\right| \leq \sum_{a'} \pi(a'|x')|Q_1(x',a') - Q_2(x',a')| \leq \max_{a'}|Q_1(x',a') - Q_2(x',a')| .$$

ii). Let us write:

$$W_1((\Pi_\mathcal{C}\mathfrak{T}\mu_1)^{(x,a)}, (\Pi_\mathcal{C}\mathfrak{T}\mu_2)^{(x,a)}) =$$

$$W_1(\sum_{x'} P(x'|x,a)\Pi_\mathcal{C}\delta_{r(x,a,x')+\gamma\max_{a'}Q_1(x',a')}, \sum_{x'} P(x'|x,a)\Pi_\mathcal{C}\delta_{r(x,a,x')+\gamma\max_{a'}Q_2(x',a')})$$

$$\leq \sum_{x'} P(x'|x,a)W_1(\Pi_\mathcal{C}\delta_{r(x,a,x')+\gamma\max_{a'}Q_1(x',a')}, \Pi_\mathcal{C}\delta_{r(x,a,x')+\gamma\max_{a'}Q_2(x',a')})$$

$$\leq \gamma \sum_{x'} P(x'|x,a)|\max_{a'}Q_1(x',a') - \max_{a'}Q_2(x',a')|$$

$$\leq \gamma \sum_{x'} P(x'|x,a)\max_{a'}|Q_1(x',a') - Q_2(x',a')| = \gamma \sum_{x'} P(x'|x,a)\max_{a'}\left|\int_{\tau=0}^1 (F_{x',a'}^{-1}(\tau) - \tilde{F}_{x',a'}^{-1}(\tau))d\tau\right|$$

$$\leq \gamma \sum_{x'} P(x'|x,a)\max_{a'}\underbrace{\int_{\tau=0}^1 |F_{x',a'}^{-1}(\tau) - \tilde{F}_{x',a'}^{-1}(\tau)|d\tau}_{W_1(\mu_1^{(x',a')}, \mu_2^{(x',a')})} \leq \gamma\overline{W}_1(\mu_1,\mu_2),$$

where the second inequality follows from Lemma A.1. Taking the supremum over all $(x,a)$ concludes the proof. The proof for $\Pi_\mathcal{C}\mathfrak{T}^\pi$ is similar.

**Lemma A.1.** *Let $a, b$ be two real numbers. Then,*

$$W_1(\Pi_\mathcal{C}\delta_a, \Pi_\mathcal{C}\delta_b) \leq |a - b|.$$

*Proof.* We proceed by exhaustion of all possible (redundant) cases, up to permuting $a$ and $b$:

(i) $a, b$ are both outside the interval $(z_1, z_K]$: the proof is trivial in this case

(ii) $a \in (z_1, z_K]$, $b \in \{z_1, \ldots, z_K\}$ and $a \leq b$

(iii) $a \in (z_1, z_K]$, $b \in \{z_1, \ldots, z_K\}$ and $a > b$

(iv) $a \in (z_1, z_K]$ and $b \notin (z_1, z_K]$

(v) $a, b$ are both inside the interval $(z_1, z_K]$

Proof in case ii). We assume that $b$ belongs to the support, $b \geq a$ and $a$ lies inside the interval $(z_1, z_K]$:

$$z_j < a \leq z_{j+1} \quad \text{and} \quad b = z_{k+1} \quad \text{with} \quad 1 \leq j \leq k \leq K .$$

Then, we have:

$$
\begin{aligned}
W_1(\Pi_{\mathcal{C}}\delta_a, \Pi_{\mathcal{C}}\delta_b) &= \frac{z_{j+1} - a}{z_{j+1} - z_j}(z_{k+1} - z_j) + \frac{a - z_j}{z_{j+1} - z_j}(z_{k+1} - z_{j+1}) \\
&= (z_{k+1} - z_j)\frac{z_{j+1} - a + a - z_j}{z_{j+1} - z_j} - (z_{j+1} - z_j)\frac{a - z_j}{z_{j+1} - z_j} \\
&= z_{k+1} - z_j - (a - z_j) = z_{k+1} - a = |a - b|.
\end{aligned}
$$

Proof in case iii). Similar to (ii) except that we have $b = z_k$ with $j \geq k$:

$$
\begin{aligned}
W_1(\Pi_{\mathcal{C}}\delta_a, \Pi_{\mathcal{C}}\delta_b) &= \frac{z_{j+1} - a}{z_{j+1} - z_j}(z_j - z_k) + \frac{a - z_j}{z_{j+1} - z_j}(z_{j+1} - z_k) \\
&= (z_j - z_k)\frac{z_{j+1} - a + a - z_j}{z_{j+1} - z_j} + (z_{j+1} - z_j)\frac{a - z_j}{z_{j+1} - z_j} \\
&= z_j - z_k + a - z_j = a - z_k = |a - b|.
\end{aligned}
$$

Proof in case iv). Now, if $b$ is outside the interval $(z_1, z_k]$, say $b \leq z_1$, we deduce from the previous computation that $W_1(\Pi_{\mathcal{C}}\delta_a, \Pi_{\mathcal{C}}\delta_b) = a - z_1 \leq |a - b|$.

Proof in case v). Finally, if both $a$ and $b$ lie in $(z_1, z_K]$,

$$
\begin{aligned}
W_1(\Pi_{\mathcal{C}}\delta_a, \Pi_{\mathcal{C}}\delta_b) &= W_1\Big(\frac{z_{j+1} - a}{z_{j+1} - z_j}\delta_{z_j} + \frac{a - z_j}{z_{j+1} - z_j}\delta_{z_{j+1}}, \Pi_{\mathcal{C}}\delta_b\Big) \\
&\leq \frac{z_{j+1} - a}{z_{j+1} - z_j}W_1(\delta_{z_j}, \Pi_{\mathcal{C}}\delta_b) + \frac{a - z_j}{z_{j+1} - z_j}W_1(\delta_{z_{j+1}}, \Pi_{\mathcal{C}}\delta_b) \\
&= \frac{z_{j+1} - a}{z_{j+1} - z_j}|b - z_j| + \frac{a - z_j}{z_{j+1} - z_j}|b - z_{j+1}|, \quad (3)
\end{aligned}
$$

where we used cases (ii) and (iii) in the last equality. Then, if $b \geq z_{j+1}$, Eq. equation 3 implies

$$
\begin{aligned}
W_1(\Pi_{\mathcal{C}}\delta_a, \Pi_{\mathcal{C}}\delta_b) &\leq \frac{z_{j+1} - a}{z_{j+1} - z_j}(b - z_j) + \frac{a - z_j}{z_{j+1} - z_j}(b - z_{j+1}) \\
&= (b - z_j)\frac{z_{j+1} - a + a - z_j}{z_{j+1} - z_j} - (z_{j+1} - z_j)\frac{a - z_j}{z_{j+1} - z_j} = b - z_j - a + z_j = b - a.
\end{aligned}
$$

Symmetrically, if $b \leq z_j$, Eq. equation 3 implies

$$
W_1(\Pi_{\mathcal{C}}\delta_a, \Pi_{\mathcal{C}}\delta_b) \leq \frac{z_{j+1} - a}{z_{j+1} - z_j}(z_j - b) + \frac{a - z_j}{z_{j+1} - z_j}(z_{j+1} - b) = a - b.
$$

Lastly, if $a, b$ both belong to the same segment $(z_j, z_{j+1}]$, and say $a \leq b$, we have by direct computation:

$$
W_1(\Pi_{\mathcal{C}}\delta_a, \Pi_{\mathcal{C}}\delta_b) = (z_{j+1} - z_j)\frac{z_{j+1} - a - (z_{j+1} - b)}{z_{j+1} - z_j} = b - a .
$$

$\square$

## B   Proof of Proposition 3.2

Given the completeness of the Wasserstein space (see Bolley (2008)), by combining Proposition 3.1 with Banach's fixed point theorem, we deduce the existence and uniqueness of the fixed points.

For $(i)$, we verify:

$$(\mathfrak{T}\nu_\pi)^{(x,a)} = \sum_{x'} P(x'|x,a)\delta_{r(x,a,x')+\gamma\sum_{a'}\pi(a'|x')\sum_{x''}P(x''|x',a')(r(x',a',x'')+\gamma V^\pi(x''))}$$
$$= \sum_{x'} P(x'|x,a)\delta_{r(x,a,x')+\gamma\sum_{a'}\pi(a'|x')Q^\pi(x',a')} = \nu_\pi^{(x,a)}\,.$$

ii). For the control case,

$$(\mathfrak{T}\nu_*)^{(x,a)} = \sum_{x'} P(x'|x,a)\delta_{r(x,a,x')+\gamma\max_{a'}\sum_{x''}P(x''|x',a')(r(x',a',x'')+\gamma V^*(x''))}$$
$$= \sum_{x'} P(x'|x,a)\delta_{r(x,a,x')+\gamma\max_{a'}Q^*(x',a')} = \nu_*^{(x,a)}\,.$$

iii). We start by computing:

$$(\mathfrak{T}^\pi \circ \Pi_\mathcal{C}\nu_\pi)^{(x,a)} = \sum_{x'} P(x'|x,a)\delta_{r(x,a,x')+\gamma\sum_{a'}\pi(a'|x')Q^\pi(x',a')} = \nu_\pi^{(x,a)}\,,$$

where we used the *mean-preserving property* of the Cramér projection (see Eq. equation 2). Then, we deduce that

$$\Pi_\mathcal{C} \circ \mathfrak{T}^\pi \circ \Pi_\mathcal{C}\nu_\pi = \Pi_\mathcal{C}\nu_\pi = \eta_\pi\,.$$

iv). Similarly, one can show that $\eta_*$ is the fixed point of $\Pi_\mathcal{C} \circ \mathfrak{T}$.

## C   Proof of Theorem 4.1

We follow the same steps as in the proofs of Theorem 2 in (Tsitsiklis, 1994) and Theorem 1 in (Rowland et al., 2018). We focus on the control case (ii), though the proof remains similar for policy evaluation. Let us define for each $(x,a) \in \mathcal{X} \times \mathcal{A}$: $L_0^{(x,a)} = \delta_{z_1}$, $U_0^{(x,a)} = \delta_{z_K}$, and for $j \geq 0$,

$$L_{j+1}^{(x,a)} = \frac{1}{2}L_j^{(x,a)} + \frac{1}{2}(\Pi_\mathcal{C}\mathfrak{T}L_j)^{(x,a)} \quad \text{and} \quad U_{j+1}^{(x,a)} = \frac{1}{2}U_j^{(x,a)} + \frac{1}{2}(\Pi_\mathcal{C}\mathfrak{T}U_j)^{(x,a)}\,.$$

We now state the two following lemmas and then, before proving them, explain how they allow us to conclude.

**Lemma C.1.**   *(i) In terms of entrywise stochastic dominance, it holds for all $j \geq 0$: $L_{j+1} \geq L_j$ and $U_{j+1} \leq U_j$.*

*(ii) $(L_j)$ and $(U_j)$ both converge to $\eta_*$ in $\overline{W}_1$.*

**Lemma C.2.**  *Given $j \geq 0$, there exists a random time $T_j \geq 0$ such that*

$$L_j \leq \eta_t \leq U_j \quad \text{for all } t \geq T_j\,, \quad \text{almost surely.}$$

From Lemma C.1-(ii), let $\epsilon > 0$ and take $j \geq 0$ large enough such that

$$\max\{\overline{W}_1(L_j, \eta_*), \overline{W}_1(U_j, \eta_*)\} < \epsilon\,.$$

Then, it follows from Lemma C.2 followed by a triangular inequality that

$$\overline{W}_1(\eta_t, L_j) \leq \overline{W}_1(L_j, U_j) \leq \overline{W}_1(L_j, \eta_*) + \overline{W}_1(U_j, \eta_*) < 2\epsilon\,,$$

for all $t \geq T_j$, almost surely. Finally,

$$\overline{W}_1(\eta_t, \eta_*) \leq \overline{W}_1(\eta_t, L_j) + \overline{W}_1(L_j, U_j) + \overline{W}_1(U_j, \eta_*) < 5\epsilon\,,$$

which implies Theorem 4.1. We still need to prove Lemmas C.1-C.2.

### C.1 Proof of Lemma C.1

(i). Let us show by induction that $U_{j+1} \leq U_j$. First, the base case $U_1^{(x,a)} \leq U_0^{(x,a)} = \delta_{z_K}$ is true because $U_1^{(x,a)}$ is supported in $\{z_1, \ldots, z_K\}$. Now, let us assume that $U_{j+1} \leq U_j$ for some $j \geq 0$. We recall from Proposition 5 in (Rowland et al., 2018) that $\Pi_{\mathcal{C}}$ is a monotone map for element-wise stochastic dominance. Plus, it is easy to see that both $\mathcal{T}^\pi$ and $\mathcal{T}$ are monotone too. Then, by monotonicity of $\Pi_{\mathcal{C}}\mathcal{T}$, it holds that $\Pi_{\mathcal{C}}\mathcal{T}U_{j+1} \leq \Pi_{\mathcal{C}}\mathcal{T}U_j$ and hence,

$$U_{j+2}^{(x,a)} = \frac{1}{2}U_{j+1}^{(x,a)} + \frac{1}{2}(\Pi_{\mathcal{C}}\mathcal{T}U_{j+1})^{(x,a)} \leq \frac{1}{2}U_j^{(x,a)} + \frac{1}{2}(\Pi_{\mathcal{C}}\mathcal{T}U_j)^{(x,a)} = U_{j+1}^{(x,a)},$$

which proves the induction step. Symmetrically, one can show that $L_{j+1} \geq L_j$.

(ii). Similarly to Lemma 7 in (Rowland et al., 2018), we formulate the following general result.

**Lemma C.3.** *Let* $(\nu_k)_{k=0}^\infty$ *be a sequence of probability distributions over* $\{z_1, \ldots, z_K\}$ *such that* $\nu_{k+1} \leq \nu_k$ *for all* $k \geq 0$. *Then, there exists a limit probability distribution* $\nu_{\lim}$ *over* $\{z_1, \ldots, z_K\}$ *such that* $\nu_k \to \nu_{\lim}$ *in* $W_1$.

*Proof.* Let us denote $F_k$ the CDF of $\nu_k$ and $F_k^{-1}$ its quantile function valued in $\{z_1, \ldots, z_K\}$. The assumption that $\nu_k$ stochastically dominates $\nu_{k+1}$ reformulates as $F_{k+1}^{-1}(\tau) \leq F_k^{-1}(\tau)$ for all $0 < \tau \leq 1$. Hence for each $\tau \in (0, 1]$, the sequence $F_k^{-1}(\tau)$ is non-increasing and lower bounded by $z_1$. Therefore, this sequence converges: $F_k^{-1}(\tau) \to H_{\lim}(\tau)$. As $F_k^{-1}(\tau)$ can only take discrete values, there exists $N(\tau)$ such that $\forall k \geq N(\tau)$, $F_k^{-1}(\tau) = H_{\lim}(\tau)$. The limit function $H_{\lim}$ is thus non-decreasing, valued in $\{z_1, \ldots, z_K\}$ and right-continuous with left limits. It is therefore the quantile function of a probability distribution $\nu_{\lim}$ supported over $\{z_1, \ldots, z_K\}$. Let us now show that $W_1(\nu_k, \nu_{\lim}) \to 0$. Denoting $F_{\lim}$ the CDF of $\nu_{\lim}$ and $p_0 = 0, p_1 = F_{\lim}(z_1), \ldots, p_K = F_{\lim}(z_K) = 1$, we have:

$$W_1(\nu_k, \nu_{\lim}) = \int_{\tau=0}^1 |F_k^{-1}(\tau) - H_{\lim}(\tau)|d\tau = \sum_{1 \leq k \leq K : p_k \neq p_{k-1}} \int_{\tau=p_{k-1}}^{p_k} |F_k^{-1}(\tau) - z_k|d\tau. \tag{4}$$

Let $\Delta_p = \min_{1 \leq k \leq K : p_k \neq p_{k-1}}(p_k - p_{k-1})$, $\Delta_z = z_K - z_1$ and $0 < \epsilon < \Delta_p/2$. Then for all $k \geq \max_{1 \leq k \leq K : p_k \neq p_{k-1}} \max\{N(p_{k-1} + \epsilon), N(p_k - \epsilon)\}$, $F_k^{-1}(\tau)$ is constant equal to $z_k$ for any $\tau \in [p_{k-1} + \epsilon, p_k - \epsilon]$ and we have:

$$W_1(\nu_k, \nu_{\lim}) = \sum_{1 \leq k \leq K : p_k \neq p_{k-1}} \int_{\tau=p_{k-1}}^{p_k} |F_k^{-1}(\tau) - z_k|d\tau$$

$$= \sum_{1 \leq k \leq K : p_k \neq p_{k-1}} \int_{\tau=p_{k-1}}^{p_{k-1}+\epsilon} \underbrace{|F_k^{-1}(\tau) - z_k|}_{\leq \Delta_z} d\tau + \underbrace{\int_{\tau=p_{k-1}+\epsilon}^{p_k-\epsilon} |F_k^{-1}(\tau) - z_k|d\tau}_{=0} + \int_{\tau=p_k-\epsilon}^{p_k} \underbrace{|F_k^{-1}(\tau) - z_k|}_{\leq \Delta_z} d\tau$$

$$\leq 2K\Delta_z\epsilon,$$

which proves the result.

$\square$

By applying Lemma C.3 to the sequence $(U_k^{(x,a)})_{k \geq 0}$ for each pair $(x, a)$, we deduce the convergence of $(U_k)$ towards some limit distribution function $U_{\lim}$ in $\overline{W}_1$. Finally, by continuity of $\Pi_{\mathcal{C}}\mathcal{T}$ for the metric $\overline{W}_1$, this limit must verify:

$$U_{\lim} = \frac{1}{2}U_{\lim} + \frac{1}{2}\Pi_{\mathcal{C}}\mathcal{T}U_{\lim} \implies U_{\lim} = \Pi_{\mathcal{C}}\mathcal{T}U_{\lim},$$

from which we deduce that $U_{\lim} = \eta_*$ by unicity. The proof that $(L_j)$ converges to $\eta_*$ in $\overline{W}_1$ is analogous.

## C.2 Proof of Lemma C.2

Let us prove Lemma C.2 by induction. The base case $j = 0$ is true because for all $t \geq T_0 = 0$, $\eta_t^{(x,a)}$ is supported on $\{z_1, \ldots, z_K\}$ and so:

$$L_0^{(x,a)} = \delta_{z_1} \leq \eta_t^{(x,a)} \leq \delta_{z_K} = U_0^{(x,a)} \ .$$

Then, let us assume that the result is true for some $j \geq 0$, i.e. there exists a random time $T_j$ such that $L_j \leq \eta_t \leq U_j$ for all $t \geq T_j$ almost surely. Now, let us show that there exists a random time $T_{j+1}$ such that $\eta_t \leq U_{j+1}$ for all $t \geq T_{j+1}$ almost surely ( the proof for $\eta_t \geq L_{j+1}$ is analogous). For each state-action pair $(x, a)$, we define $H_{T_j}^{(x,a)} = U_j^{(x,a)}$ and $W_{T_j}^{(x,a)}$ the zero measure i.e. $W_{T_j}^{(x,a)}(A) = 0$ for all Borel sets $A \subseteq \mathbb{R}$. Then, for $t \geq T_j$,

$$H_{t+1}^{(x,a)} = (1 - \alpha_t(x,a))H_t^{(x,a)} + \alpha_t(x,a)(\Pi_{\mathcal{C}}\mathcal{T}U_j)^{(x,a)}$$
$$\text{and} \qquad W_{t+1}^{(x,a)} = (1 - \alpha_t(x,a))W_t^{(x,a)} + \alpha_t(x,a)[\Pi_{\mathcal{C}}(\delta_{r(x,a,x')+\gamma \max_{a'} Q_t(x',a')}) - (\Pi_{\mathcal{C}}\mathcal{T}\eta_t)^{(x,a)}] \ , \quad (5)$$

where $x' \sim P(\cdot|x,a)$, $Q_t(x',a') = \sum_{k=1}^K p_{t,k}(x',a')z_k$. Moreover, if $(x,a) = (x_t,a_t)$, then $x' = x_{t+1}$ and $r(x_t, a_t, x_{t+1}) = r_t$. A consequence of Eq. (5) is that $W_t^{(x,a)}$ is a signed measure on $\mathbb{R}$ with total measure equal to zero: $W_t^{(x,a)}(\mathbb{R}) = 0$ for all $t \geq T_j$. Let us now prove by (another) induction that $\eta_t^{(x,a)} \leq H_t^{(x,a)} + W_t^{(x,a)}$ for all $t \geq T_j, (x,a) \in \mathcal{X} \times \mathcal{A}$ almost surely. For $t = T_j$, it is true by assumption that $\eta_{T_j} \leq U_j$ almost surely. Then, suppose that $\eta_t \leq H_t + W_t$ almost surely for some $t \geq T_j$. Recalling that $\alpha_t(x,a) = 0$ for all $(x,a) \neq (x_t,a_t)$, it holds that:

$$\eta_{t+1}^{(x,a)} = (1 - \alpha_t(x,a))\eta_t^{(x,a)} + \alpha_t(x,a)\Pi_{\mathcal{C}}(\delta_{r(x,a,x')+\gamma \max_{a'} Q_t(x',a')})$$
$$= (1 - \alpha_t(x,a))\eta_t^{(x,a)} + \alpha_t(x,a)(\Pi_{\mathcal{C}}\mathcal{T}\eta_t)^{(x,a)} + \alpha_t(x,a)[\Pi_{\mathcal{C}}(\delta_{r(x,a,x')+\gamma \max_{a'} Q_t(x',a')}) - (\Pi_{\mathcal{C}}\mathcal{T}\eta_t)^{(x,a)}]$$
$$\leq (1 - \alpha_t(x,a))(H_t + W_t)^{(x,a)} + \alpha_t(x,a)(\Pi_{\mathcal{C}}\mathcal{T}U_j)^{(x,a)} + \alpha_t(x,a)[\Pi_{\mathcal{C}}(\delta_{r(x,a,x')+\gamma \max_{a'} Q_t(x',a')}) - (\Pi_{\mathcal{C}}\mathcal{T}\eta_t)^{(x,a)}]$$
$$= H_{t+1}^{(x,a)} + W_{t+1}^{(x,a)} \ , \quad (6)$$

where the inequality comes from the assumptions $\eta_t \leq H_t + W_t$ and $\eta_t \leq U_j$ combined with the monotonicity of $\Pi_{\mathcal{C}}\mathcal{T}$. Now, observe that $H_t$ can be explicitly expressed as follows:

$$H_t^{(x,a)} = \left(\prod_{t'=T_j}^{t-1}(1 - \alpha_{t'}(x,a))\right)U_j + \left(1 - \prod_{t'=T_j}^{t-1}(1 - \alpha_{t'}(x,a))\right)(\Pi_{\mathcal{C}}\mathcal{T}U_j)^{(x,a)} \ .$$

Since by Assumption 4.1, it holds that $\sum_{t'=0}^{\infty}\alpha_{t'}(x,a) = \infty$ for all $(x,a)$ almost surely, we deduce that there exists a random time $\widetilde{T}_{j+1} \geq T_j$ such that $\prod_{t'=T_j}^{t-1}(1 - \alpha_{t'}(x,a)) \leq \frac{1}{4}$ for all $(x,a)$ and for all $t \geq \widetilde{T}_{j+1}$ almost surely. Then, because Lemma C.1-(i) implies that $\Pi_{\mathcal{C}}\mathcal{T}U_j \leq U_j$, we have for all $t \geq \widetilde{T}_{j+1}$:

$$\eta_t \leq H_t + W_t \leq \frac{1}{4}U_j + \frac{3}{4}\Pi_{\mathcal{C}}\mathcal{T}U_j + W_t = \frac{1}{2}U_j + \frac{1}{2}\Pi_{\mathcal{C}}\mathcal{T}U_j + W_t - \frac{1}{4}(U_j - \Pi_{\mathcal{C}}\mathcal{T}U_j)$$
$$= U_{j+1} + W_t - \frac{1}{4}(U_j - \Pi_{\mathcal{C}}\mathcal{T}U_j) \ . \quad (7)$$

We point out that the random noise term appearing in the definition of $W_{t+1}^{(x,a)}$ has zero mean: for all $1 \leq k \leq K$,

$$\mathbb{E}_{x' \sim P(\cdot|x,a)}\left[\Pi_{\mathcal{C}}(\delta_{r(x,a,x')+\gamma \max_{a'} Q_t(x',a')}) - (\Pi_{\mathcal{C}}\mathcal{T}\eta_t)^{(x,a)}\right]((-\infty, z_k]) = 0 \ . \quad (8)$$

Eq. (8) implies via a classic stochastic approximation argument under Assumption 4.1 that $W_t^{(x,a)}((-\infty, z_k]) \to 0$ almost surely, for all $(x,a)$ and $k \in \{1, \ldots, K\}$. Finally, we take $T_{j+1} \geq \widetilde{T}_{j+1}$

large enough so that $|W_t^{(x,a)}((-\infty, z_k])| \leq \frac{\Delta}{4}$ for all $t \geq T_{j+1}$ and all $(x,a)$, almost surely, where $\Delta$ is given by:

$$\Delta = \inf \left\{ \left| \left[ U_j^{(x,a)} - (\Pi_{\mathcal{C}} \mathcal{T} U_j)^{(x,a)} \right] ((-\infty, z_k]) \right| \neq 0 : x \in \mathcal{X}, a \in \mathcal{A}, 1 \leq k \leq K \right\},$$

which concludes the proof by using Eq. (7). Indeed, if $U_j^{(x,a)}((-\infty, z_k]) = (\Pi_{\mathcal{C}} \mathcal{T} U_j)^{(x,a)}((-\infty, z_k])$, then $U_j^{(x,a)}((-\infty, z_k]) = U_{j+1}^{(x,a)}((-\infty, z_k])$.

