# OpenReview forum: "One-Step Distributional Reinforcement Learning"
_TMLR — Accepted by TMLR_

### Review · Reviewer_J5Ja · 2023-06-14

**Summary Of Contributions:**

This paper introduces a new approach for distributional RL (DistRL) in which the target distribution is modelled by a Dirac of its average (instead of the learned distribution at the next step in classical DistRL).

That simplification makes the (cramér-projected) Bellmann operator contractive and solves the instability issue in case of non-unique optimal policy.


**Audience:**

Yes

**Broader Impact Concerns:**

Does not apply

**Claims And Evidence:**

Yes

**Requested Changes:**

Critical changes
------------------

The paper needs more experimental justification. At least averaging over 5 seeds and much more games (the ideal would be all the 57 Atari games).

Other changes
-----------------

One important ingredient for the Banach's fixed point theorem is that the metric space is complete, which is never specified.
In the case of Wasserstein metrics on probability distributions the completeness is known (for example: Bolley, François. "Separability and completeness for the Wasserstein distance.") but I think a sentence should mention that as it is not a well-known or trivial result.

In def 3.1 (and 3.2 and next) it took me a while to understand that all the target terms are in the Dirac (and not just the reward). I guess just adding parenthesis would help the reader a lot, or even a notation \delta(x) instead of \delta_{x}.

In fig. 4 (Frozen Lac) it would be great to explain a bit what are theses states and what the different actions mean.

In Appendix proof of prop1, I did not understand the sentence
" where the first inequality follows from the interpretation of  the Wassertein distance as an infimum over couplings "
Isn't the first inequality a simple Jensen (W_p being convex) ?



**Strengths And Weaknesses:**

Strength
---------

The trick is simple, does not affect the optimality and looks even simpler to implement than classical DistRL.

The math are solid, and making the projected Bellman operator contractive is a strong improvement.

They show a very simple example of 2-states MDP where the classical approaches fail while that one converges.

Weaknesses
--------------

The experimental part looks a bit sloppy, with only 3 games and results averaged over 3 seeds.

This makes the result very hard to interpret and I would not say that the new method does not deteriorate the performance of classical DistRL.

---

> ### Author Response · Authors · 2023-06-22
> **Atari experiments on two additional seeds**
>
> Dear Reviewer,
>
> We greatly appreciate your constructive feedback and recognition of the theoretical contributions of our new approach to DistrRL.
>
> 1. **Experiments**: We acknowledge your concerns about the scope of our experimental data. We are currently in the process of running two additional seeds for all five algorithms across the same three games. Unfortunately, due to constraints in computing resources, we are unable to conduct experiments on a larger set of Atari games. Given the theoretical nature of our primary contribution, we believe that the experiments carried out on five distinct environments (namely, a toy MDP, Frozen Lake, and three different Atari games) provide sufficient evidence to back our theoretical contributions. We concur with your observation that one-step DistrRL appears slightly inferior to the original DistrRL on Atari games; this could be due to the fact that full DistrRL learns a richer distribution compared to the one-step method. We will add a discussion to this effect in our revised manuscript. However, we note that our approach scales well in a deep RL setting, obtaining better results than DQN.
>
> 2. **Completeness of the Wasserstein Space**: We are grateful for your reference suggestion, and we agree that it is crucial to recall the completeness of the Wasserstein space. We will include the citation in our revised manuscript.
>
> 3. **Frozen Lake Description**: We will include a brief explanation detailing the states and actions in the revised manuscript.
>
> 4. **Jensen Argument**: We understand your comment concerning the Jensen inequality. However we did not find any reference with a simple/clear statement of the joint convexity of $W_p(\cdot, \cdot)$ (except the "partition lemma" in Bellemare et al. 2017, whose proof is a bit involved). To circumvent any ambiguity, we have simply upper bounded $W_p^p( T \mu_1, T\mu_2 )$ via a specific coupling, $\lambda$, such that $$\lambda( \\{ r(x,a,x')+\gamma \max_{a'} Q_1(x',a') \\} \times \\{ r(x,a,x')+\gamma \max_{a'} Q_2(x',a') \\} ) = P(x'|x,a)$$. This argument will be clarified in the revised manuscript.
>
> We are thankful for your suggestions and will incorporate them to further improve our paper.

---

### Review · Reviewer_Q3pB · 2023-06-15

**Summary Of Contributions:**

This paper proposes two new distributional reinforcement learning (RL) algorithms and studies their convergence properties. The two algorithms include a prediction algorithm and a control algorithm. The main contribution is that the new control algorithm can be shown to converge while the existing distributional control algorithms can not. Technically, this contribution is not novel because it mainly uses tools used in previous works. To further verify this convergence result, this paper also shows empirically that the control algorithm indeed converges in a small-scale environment called Frozen Lake environment, thus verifying the convergence result. In addition, the neural network extension of the control algorithm has been tested in three commonly-used atari domains. The empirical results of the new control algorithm seem to be comparable with a state-of-the-art distributional control algorithm.


**Audience:**

Yes

**Claims And Evidence:**

Yes

**Requested Changes:**

see weaknesses

**Strengths And Weaknesses:**

positive points:

The overall writing is clear.

The technical results look solid to me.

The claim of contribution is clear and the results support the claim well.

negative points:

The paper does not clearly studies what we lose by considering the one-step distribution instead of the full return distribution. To appreciate the importance of the new algorithms, this point needs to be clearly said.

The paper does not provide insights into why existing distributional control algorithms fail to converge and how the new algorithm overcomes this issue.

The empirical results lack experiment details (like what hyperparameters have been tested and how the curves in the figures are generated).

---

> ### Author Response · Authors · 2023-06-19
> **Clarifications on One-Step Distributional Reinforcement Learning and Experiment Parameters**
>
> Dear Reviewer,
>
> We greatly appreciate your time and constructive feedback on our work.
>
> We apologize for any lack of clarity and aim to address your concerns, many of which we believe have been covered within the body of our work.
>
> When comparing our one-step method to traditional/full Distributional Reinforcement Learning (DistrRL), our method learns a different and simpler fixed point. This is illustrated by Figure 3 and Table 1. Unlike the continuous distribution typically achieved by full DistrRL, our one-step fixed point is an atomic distribution. We will be adding additional commentary in the paper to clarify this point further.
>
> The instability and failure of full DistrRL to converge, especially when there are numerous optimal policies, is explored in Subsection 2.3. We cite Propositions 1-2 from Bellemare et al. 2017 which provide theoretical arguments supporting this phenomenon. The convergence of our method is achieved because the fixed point is a mixture of optimal values $V^*(x')$ in next states and remains consistent for any optimal policy. We realize that the brief explanation at page 8, "Contrary to the original CDRL target [...] as illustrated in Figure 2", may need additional clarification and we will enhance this explanation.
>
> Concerning the experimental details, we used the default parameters from the standard "CleanRL" codebase for each algorithm. In order to improve the clarity, we will add these parameters to the paper. For our new OC-C51 method, we have used the same default parameters as in C51 due to their similarities.
>
> Again, we thank you for your insightful review and hope that these explanations address your concerns. We are confident that with these clarifications, our paper can offer significant value to the research community.

---

### Review · Reviewer_SmEz · 2023-06-19

**Summary Of Contributions:**

This paper presents a variant of distributional reinforcement learning (DistRL), where the standard Bellman backup is replaced by a one-step backup. The advantage of this approach is it allows proving convergence in both the policy evaluation and control cases, without requiring assumptions such as uniqueness of the optimal policy (as required in prior work). Through empirical evaluations the authors also present evidence that their simpler method can reduce training instabilities in certain edge cases, and has comparable performance in some standard benchmark environments.

**Audience:**

Yes

**Claims And Evidence:**

Yes

**Requested Changes:**

The main point to be addressed are:
1. The one pointed out above, with regards to the use of value functions in the theoretical analyses. This is a critical point to address.
1. The conclusion is a bit terse. I think it could be expanded a bit more, in particular with respect to limitations, practical implications, and possibilities for future work.
1. In the proof of Lemma A.1 the second inequality for case ii doesn't seem to follow. The first line expands to
$z_{j+1}z_{k+1} - z_{j+1}z_j - az_{k+1} + az_j + az_{k+1} - az_{j+1} - z_j z_{k+1} + z_j z_{j+1}$

whereas the second line ends up with an extra $-z_j^2$ term.

Some other minor points:
1. Figure 1 could be moved further down, it's a little distracting where it is, and not needed there.
1. You should cite the recent [Distributional RL book](https://www.distributional-rl.org/), by Bellemare, Dabney, and Rowland. In particular, there's a counterexample in chapter 7 showing that one can't remove the condition of unique optimal policy to guarantee convergence in the regular DistRL setting.
1. In Figure 3:

    a. Add a legend

    b. Add the discussion on growing number of atoms in the caption

    c. Remind the user the difference between $\mathcal{T}$ and $\mathscr{T}$
1. In Algorithm 1, where are the $p_{t,k}$ updated? The $t$ subscript suggests they should also be updated in the loop, but it is not specified.
1. In Figure 4 are you only showing OS-CDRL? How does CDRL compare?
1. Add more context to the caption of figure 4; it's best if figures are (mostly) self-contained.
1. Why were $\gamma=1/2$ and specific categories chosen for the DP example in 5.1? Are the results sensitive to this?
1. In the DP example in 5.1, could one argue that the increased number of atoms results in more expressivity of the distribution, which is necessary for regular DistRL, but not necessary for one-step? Put another way, is it just that full DistRL results fin a richer distribution that requires more atoms?
1. In the Frozen lake discussion you seem to be suggesting a distributional approach is not justified in deterministic settings. Is this necessarily the case?
1. In 5.2 are the same hyperparameters used for C51?
1. In 5.2 are you using sticky actions?
1. Please restate proposition/theorem statements in appendix

**Strengths And Weaknesses:**

# Strengths

The paper is very well written. I particularly liked section 2 and the discussion in section 4.1.
I also appreciate the motivation for this work. Even though it's a simplification of the original DistRL algorithm, this enables theoretical results that were previously not possible (although see questions below).
The empirical evaluations are also effective at transmitting the message the authors are trying to convey.

# Weaknesses

## Source of value functions
My main concern with this paper is around Proposition 3.2, which has implications for Theorem 4.1, the main theoretical result of the paper.
In the statement of proposition 3.2 the authors define $\nu_{\pi}^{(x, a)}$ and $\nu_{\*}^{(x, a)}$ with respect to $V^{\pi}(x')$ and $V^*(x')$, respectively (also mentioned in Table 1). Where are $V^{\pi}$ and $V^*$ coming from? Are they the value functions you'd obtain with (standard) expectational RL? If so, this appears to be problematic, as they would _not_ be the same values you'd get with the proposed one-step method. Indeed, the proof of Proposition 3.2 seems to rely on $\nu$ being defined with respect to these value functions.

Going through the proof, it seems like the value functions can basically be replaced with _any_ function $f:\mathcal{X}\rightarrow\mathbb{R}$, as long as $\nu$ is defined with respect to $f$ in the proposition statement. In other words, there is no need for the true value functions for the proof to hold.

And this leads me to question the significance of the results. It seems they're really mostly concerned with immediate reward, but are basically agnostic to downstream implications that are affected by the environment (see comment above about replacing with any state function). This makes sense, since it is one-step after all, but then I'm not sure on the claimed strength of connection to the original distributional RL results.

The empirical results suggest otherwise, but I'm wondering if this is a consequence of the environments rather than the proposed algorithm. A baseline that would be more convincing is to run a one-step DQN, say. This would verify if dropping the Bellman backup still results in good performance. If so, then I would search for environments where this is _not_ the case. Also consider adding sticky actions (if you're not using them already).

---

> ### Author Response · Authors · 2023-06-29
> **Value function cannot be replaced by any function + No extra term in the proof of Lemma A.1**
>
> Dear Reviewer,
>
> We appreciate your insightful comments and the opportunity to clarify our findings.
> The minor points we haven't addressed here will be duly considered and incorporated in our revised manuscript.
>
> ## Main Points
>
> **1. Clarification of Proposition 3.2**: The proof of Proposition 3.2 (point i, for instance) involves replacing, inside each Dirac, the quantity $\sum_{x''} P(x'' | x',a')( r(x',a', x'') + \gamma V^{\pi}(x'') )$ with the simpler $Q^{\pi}(x',a')$. This substitution is permissible due to the fundamental fixed-point characterization of $V^\pi$ and $Q^\pi$, hence, these functions cannot be replaced. The misunderstanding might have arisen from our Dirac notations. The full Bellman target (reward + discounted value in the next state) is "inside" the Dirac, not just the reward. If this was unclear, we will make necessary modifications in the revised manuscript.
>
> **2. Discussion on Limitations and Future Work**: We will include an expanded conclusion in our revised manuscript, addressing the limitations of one-step fixed-point (for instance, inability to distinguish two policies with identical Q-functions but different risk-levels), and discuss future research directions on multi-step DistrRL.
>
> **3. No "extra term"**: We believe you intended "equality" instead of "inequality" (otherwise your comment does not make sense). We invite you to take another look at the second line: you will notice that the "extra term" $-z_j^2$ is effectively cancelled by an additional term $+z_j^2$.
>
> ## Minor Points
>
> **4. Updated probabilities**: $\eta_t$ and $\eta_{t+1}$ are distributions with the same support, but their probabilities, $p_{t,k}$ and $p_{t+1,k}$, differ. The updated probabilities are given by
> $p_{t+1,k} = (1-\alpha_{t}) p_{t,k} + \alpha_{t} \hat p_{t,k}$,
> with $\hat \eta_t = \sum_{k} \hat p_{t,k} \delta_{z_k}$.
> This was not explicitly stated to avoid introducing additional notation $\hat{p}_{t,k}$. This will be clarified in our revised manuscript.
>
> **5. Purpose of Figure 4**: Figure 4 was intended to demonstrate the convergence of OS-CDRL. In this example, we believe that CDRL probabilities would also converge but to different values due to their different update rule. Nevertheless, the averaged Q-function would remain identical in both scenarios due to the mean-preserving property of the Cramer projection (refer to Eq. 2).
>
> **7. Sensitivity of Results**: The results are indeed sensitive to the support choice. As seen in Figure 1's toy MDP (combined with $\gamma=1/2$ ), the action $a_1$ in state $x_1$ results in deterministic transitions and a total discounted return reduced to a Dirac at 2. Therefore, we selected a support with values 1.9 and 2.1 around the true value 2. However, the action $a_2$ leads to the same expected value $Q^*(x_1, a_2) = 2$ but with non-zero variance. Thus, CDRL targets can occasionally (when taking action $a_2$) fall outside the [1.9, 2.1] range, leading to the depicted instability.
>
> **8. DistrRL Approximation**: While full DistrRL typically attempts to approximate continuous distributions (requiring infinite atoms for perfect approximation), the fixed point in one-step DistrRL is always an atomic distribution consisting of |X| atoms (|X| is the cardinality of the state space). This suggests that fewer atoms are needed in one-step DistrRL than in full DistrRL as the full DistrRL fixed point carries more information.
>
> **9. Deterministic Control Task**: In our setting, we consider a deterministic reward function $r(x, a, x')$ and the primary source of randomness is the transition kernel $P(x'|x,a)$. If we assume that $P$ is deterministic (values only 0 or 1), then the only remaining source of randomness is the policy $\pi$ (if stochastic policies are considered). In our Algorithm 1, policy stochasticity only plays a role in the policy evaluation case and does not influence the control task.
>
> **10. Hyperparameters for OS-C51 and DQN**: The hyperparameters used for OS-C51 are identical to those used for C51, i.e., the default hyperparameters from the standard "CleanRL" GitHub repository. Similarly, we used the default hyperparameters from CleanRL for DQN. This will be specified in our revised manuscript.
>
> **11. No Sticky Actions**: We have employed the default setup from CleanRL, which does not involve sticky actions.
>
> In light of these clarifications and considering the misunderstanding in the two main points, we kindly invite you to reconsider your assessment and change the 'No' to a 'Yes' in the 'Claims and Evidence' section. We assure you that all your comments are greatly appreciated and will help us improve the quality of our manuscript.

---

> > ### Comment · Reviewer_SmEz · 2023-06-30
> > **Proposition 3.2**
> >
> > Thanks for your response, some responses below.
> >
> > 1. **Clarification of Proposition 3.2:** Thanks for your clarification, I think the subscripts with the deltas were part of my confusion. The result is somewhat counterintuitive, in that by a one-step operator you still get the fixed point as being a function of the fixed-point of the expectational operator. Although you have some discussion after proposition 3.2, I think it'd be worthwhile to expand on this a bit more, as I feel it is rather deep. Perhaps some visualizations of the resulting fixed points for some simple system could help drive the point home even further.
> > I will update my score in the 'Claims and Evidence' section.
> >
> > 2. **Discussion on Limitations and Future Work:** Great, looking forward to reading this in the next revision.
> >
> > 3. **No "extra term":** Yes, I meant equality. I'll take another look, it's possible I made a mistake when revising it. For what it's worth, the proof of case (iii) was ok when I reviewed it :).
> >
> > ## Minor points
> >
> > **4.** Great, looking forward to reading this in the next revision.
> >
> > **11.** You should be able to modify the call to `gym.make` to enable sticky actions [here](https://github.com/vwxyzjn/cleanrl/blob/6c3e3c526cacc97c42abfef101329c737a11b7fd/cleanrl/c51_atari.py#L96).
> >
> > The call would be something like `gym.make(env_id, repeat_action_probability=0.25)`. You can see the constructor parameters that you can adjust in [this file](https://github.com/mgbellemare/Arcade-Learning-Environment/blob/259f24951d27bdfcb5d7b3f54f1f420ca44b71ef/src/python/env/gym.py#L42).
> >
> > It would be great if you can run some experiments with sticky actions so as to evaluate how well OS-C51 works in this case.

---

### Comment · Reviewer_Q3pB · 2023-06-13
**Interesting paper but lacks some important discussion**

This paper proposes two new distributional reinforcement learning (RL) algorithms and studies their convergence properties. The two algorithms include a prediction algorithm and a control algorithm. The main contribution is that the new control algorithm can be shown to converge while the existing distributional control algorithms can not. Technically, this contribution is not novel because it mainly uses tools used in previous works. To further verify this convergence result, this paper also shows empirically that the control algorithm indeed converges in a small-scale environment called Frozen Lake environment, thus verifying the convergence result. In addition, the neural network extension of the control algorithm has been tested in three commonly-used atari domains. The empirical results of the new control algorithm seem to be comparable with a state-of-the-art distributional control algorithm.

positive points:

The overall writing is clear.

The technical results look solid to me.

The claim of contribution is clear and the results support the claim well.

negative points:

The paper does not clearly studies what we lose by considering the one-step distribution instead of the full return distribution. To appreciate the importance of the new algorithms, this point needs to be clearly said.

The paper does not provide insights into why existing distributional control algorithms fail to converge and how the new algorithm overcomes this issue.

The empirical results lack experiment details (like what hyperparameters have been tested and how the curves in the figures are generated).

---

> ### Author Response · Authors · 2023-06-15
> **Clarifications on One-Step Distributional Reinforcement Learning and Experiment Parameters**
>
> Dear Reviewer,
>
> We greatly appreciate your time and constructive feedback on our work.
>
> We apologize for any lack of clarity and aim to address your concerns, many of which we believe have been covered within the body of our work.
>
> When comparing our one-step method to traditional/full Distributional Reinforcement Learning (DistrRL), our method learns a different and simpler fixed point. This is illustrated by Figure 3 and Table 1. Unlike the continuous distribution typically achieved by full DistrRL, our one-step fixed point is an atomic distribution. We will be adding additional commentary in the paper to clarify this point further.
>
> The instability and failure of full DistrRL to converge, especially when there are numerous optimal policies, is explored in Subsection 2.3. We cite Propositions 1-2 from Bellemare et al. 2017 which provide theoretical arguments supporting this phenomenon. The convergence of our method is achieved because the fixed point is a mixture of optimal values $V^*(x')$ in next states and remains consistent for any optimal policy. We realize that the brief explanation at page 8, "Contrary to the original CDRL target [...] as illustrated in Figure 2", may need additional clarification and we will enhance this explanation.
>
> Concerning the experimental details, we used the default parameters from the standard "CleanRL" codebase for each algorithm. In order to improve the clarity, we will add these parameters to the paper. For our new OC-C51 method, we have used the same default parameters as in C51 due to their similarities.
>
> Again, we thank you for your insightful review and hope that these explanations address your concerns. We are confident that with these clarifications, our paper can offer significant value to the research community.

---

> > ### Comment · Reviewer_Q3pB · 2023-06-17
> > **Ask for further clarification**
> >
> > Thanks for the authors' responses.
> >
> > In terms of the comparison of the one-step and full-distribution cases, the explanation does not look complete to me. Specifically, why is an atomic distribution better? Are there other advantages of considering the one-step distribution? And what do we lose by considering the one-step distribution?
> >
> > The second answer is not clear either. The authors mentioned that "The convergence of our method is achieved because the fixed point is a mixture of optimal values in next states and remains consistent for any optimal policy." What does the fixed point being consistent for any optimal policy mean? Why does it result in convergence? What is missing in the full-distribution case so that we lose convergence?
> >
> > I am fine with using the same hyper-parameters as in an existing codebase because the experiments are not major contributions of this paper, as long as these hyper-parameters and other details like how to read the curves are explicitly said.

---

> > > ### Author Response · Authors · 2023-06-19
> > > **Comparative table**
> > >
> > > Please find below a comparison between one-step DistrRL and Full DistrRL in terms of their benefits and limitations.
> > >
> > > |                     | One-step DistrRL | Full DistrRL |
> > > |---------------------|------------------|--------------|
> > > | Pros                | Same fixed-point for all optimal policies: implies stability in the control task. | The fixed point is more informative: possible to distinguish two policies with same value functions but different distributions. |
> > > | Cons                | The fixed point is less informative: impossible to distinguish two optimal policies that yield different risk levels (e.g., different variances). | Different optimal policies can lead to different fixed points, causing potential instability as the control algorithm then keeps switching between these different optimal policies and never converges. |
> > >
> > > We will include explicit details about the hyperparameters and instructions on how to read the curves in our revised manuscript.

---

### Author Response · Authors · 2023-07-05
**Revised manuscript**

Dear Reviewers,

A revised version of our manuscript has been uploaded. This updated version includes additional discussions and clearer explanations to address the concerns raised during the review process. **The modifications in the manuscript are highlighted in red.**

Another updated version of the manuscript is in progress, including new figures for Atari experiments based on 2 additional seeds: it should be available very soon.

---

> ### Author Response · Authors · 2023-07-06
> **New plots for Atari experiments**
>
>
> Dear Reviewers,
>
> The latest version of our manuscript, including new **Atari experiment** plots with **2 additional seeds (5 seeds in total)**, is now available.

---

### Decision · Action_Editors · 2023-07-25

**Recommendation:** Accept with minor revision

**Comment:**

This paper identifies a problem in the theoretical application of the most common form of distributional RL. It proposes a new one-step rule, supported by a solid formal analysis (including a convergence guarantee without the assumption of a single optimal policy in both control and evaluation settings, along with a demonstration in a small domain) and several empirical results.

There remain several outstanding points that the authors should address:
- all minor clarifications remaining from all reviews and responses,
- the outstanding clarification to Proposition 3.2 pointed out by reviewer SmEz

I also encourage the authors to add one experiment using sticky actions to improve the paper suggest by SmEz, which would improve the empirical evaluation further.


**Audience:**

The audience is appropriate for TMLR.

**Claims And Evidence:**

The paper lives up to the claims with sufficient supporting evidence as pointed out by reviewers.